# Compensative movement ameliorates reduced efficacy of rapidly-embodied decisions in humans

Akemi Kobayashi [ID] [1,2✉] & Toshitaka Kimura [ID] [1,2✉]

Dynamic environments, such as sports, often demand rapid decision-making and motor execution. The concept of embodied decision refers to the mutual link between both processes, but little is known about how these processes are balanced under severe time constraints. We address this problem by using a baseball-like hitting paradigm with and without Go/No-go judgment; participants were required to hit (Go) a moving target in the strike area or not to hit (No-go) other targets. We found that Go/No-go judgments were effective with regard to task performance, but efficacy was lost below the time constraint of 0.5 seconds mainly due to a reduction in judgment accuracy rather than movement accuracy. However, either slowing movement initiation in Go trials or canceling the movement in progress in No-go trials improved judgment accuracy. Our findings suggest that embodied decision efficacy is limited in split-second periods, but compensation is possible by changing ongoing movement strategies.

[1] NTT Communication Science Laboratories, Nippon Telegraph and Telephone Corporation, Kanagawa, Japan. [2] These authors contributed equally: Akemi Kobayashi Toshitaka Kimura. ✉email: akemi.kobayashi.rc@hco.ntt.co.jp; toshitaka.kimura.kd@hco.ntt.co.jp

In dynamically and continuously changing environments, such as sports and driving a car, we must frequently and rapidly resolve two basic tradeoffs, that is, the speed-accuracy trade-offs (SATs) for decision-making and for motor execution. As an example, consider baseball where the batter must decide whether to hit the ball and then perform a precise swing, usually in less than a second[1]. Many studies of perceptual decisions have proposed models based on the stochastic accumulation of sensory evidence toward a decision boundary[2–7]. These models assume that better decisions take longer times to accumulate sufficient sensory information since accurate decisions typically have high boundaries to reach, which is a general description of the SAT for decision-making. Similarly, the SAT for motor execution assumes that longer movement durations are needed to achieve greater movement accuracy; well-known as Fitt's law[8–10]. These findings raise the question of how the batter can allocate the time available (less than a second) to simultaneously achieve both appropriate decisions and precise bat swing.

Recent studies have described the concept of embodied decision to assess the interaction between the decision-making process and the motor system[11–13]. In this concept, the motor system not only executes the motor plan based on the decision-making process but also inversely influences the decision-making process through action dynamics and kinematics (see also reviews[14,15]). Some studies have also investigated the optimality of time distribution of decision-making and action[16–18]. Battaglia and Schrater[16] demonstrated that when trying to reach a target indicated by a fuzzy distribution of dots within a 1200-ms time limit (the number of dots increased and the decision became easier as time passed, but the time remaining for movement decreased which decreased motor accuracy), participants could vary the viewing time and remaining movement duration to yield near-optimal performance; this suggests a tradeoff in time between visual decision and motor accuracy. Reynaud et al.[19] have revealed the trade-off between decision making and motor process according to time constraints of up to a few seconds by manipulating motor cost for a responding target in a token task. Their results indicate that fast and inaccurate decisions were often made before more demanding movements so that participants sacrificed decision making for action execution. However, most embodied decision studies have targeted relatively long periods (seconds) even though many daily actions are performed under

severe time constraints (split second) and have been the subjects of some studies[20,21].

Severe time constraints are expected to impact both decision-making and motor execution. Regarding decision-making, Owens et al.[22] assessed a baseball batting-specific Go/No-go decision responses, in which participants were required to reach the screen if the moving target passed within a certain area and not to reach if it passed outside the area. In this task, the time from target start to reach contact (Time-to-contact; TTC) was restricted to 0.4 s, 0.5 s, or 0.6 s. Result showed that the decision accuracy decreased significantly when TTC reached 0.4 s, but made no mention of the motor execution aspects such as movement duration and trajectory. Movement strategy can be also changed with time constraints. Tests on the rapid interception task support the operational timing hypothesis in the broad sense that movement onset time and movement duration co-vary with task constraints[23]. For example, with time constraint of 0.5 s or above, participants could take a strategy of changing the movement onset time according to the speed of the moving target. However, when the time constraint was <0.5 s they applied the strategy of changing movement duration rather than movement onset[24]. Visuomotor delay, which is defined as the time between the emergence of visually detectable information and the initiation of resulting motor adjustment[25], is a possible determinant of sensorimotor decision behavior. The extent of this delay has been shown to vary between 100 ms and 300 ms depending on the ongoing task, such as the online regulation of the movement[26] and the initiation of movement[27], and motor expertize[28]. While these studies indicate that short time constraints affect decision-making and motor execution, it is unclear how they are balanced under severe time constraints.

The present study examines how decision-making and motor execution/regulation are achieved under severe time constraints using a baseball-like fast-hitting paradigm with Go (Strike) or No-go (Ball) decision-making. Participants are required to hit a moving target in the strike area with a hand cursor and return the target toward a frontal goal area, but to refrain from hitting the Ball targets (Fig. 1a). The Strike targets are more centered and consequently expected to be easier to hit than the Ball targets, which provides an advantage in making Go/No-go judgments. However, this task had a trade-off between judgment and movement accuracy. The participants must balance this trade-off in a limited time so as to enhance task performance. To assess

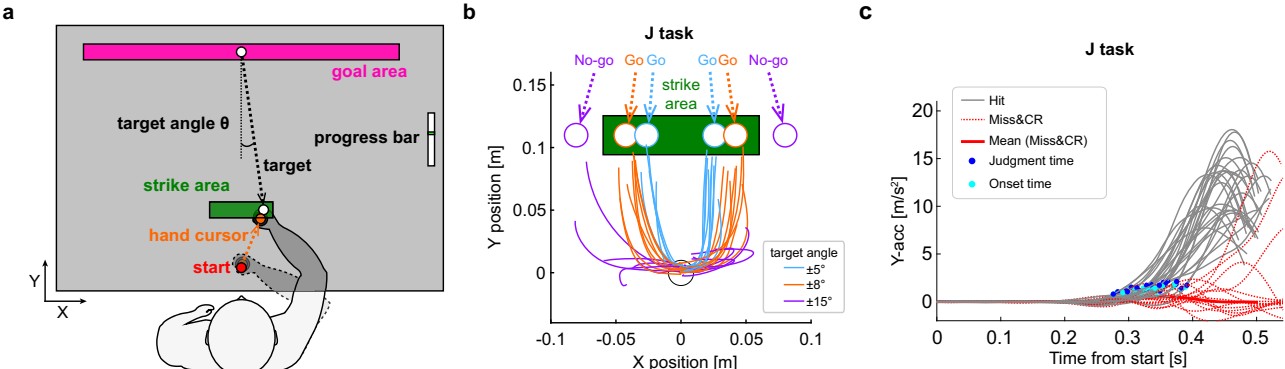

**Fig. 1 Experimental setup and typical hand trajectories. a** Schematic of the experimental setup using the manipulandum system. Participants were instructed to hit a moving target in the strike area (green square) by manipulating their hand cursor (orange circle) from the start position (red circle) so as to send the target into the goal zone (magenta square). **b** Potential target positions and typical hand trajectories for a representative participant. Targets passed through the strike area (Strike targets; ±5 (cyan), ±8° (orange)) or outside (Ball targets; ±15° (purple)) with different TTCs (0.4 s, 0.5 s, or 0.6 s). Target (dashed lines) and hand trajectories (solid lines) at 0.5 s TTC in J task are displayed as examples. **c** Typical hand acceleration profiles in the forward (y) direction at 0.5 s TTC for a representative participant. Judgment times (blue circles) were defined as the time at which each Hit trial acceleration (gray line) diverged from the average acceleration for all No-go (CR and Miss) trials (thick red line) in J task. Each CR or Miss trial is indicated by a dashed red line. Movement onset times are shown by cyan circles.

judgment efficacy, participants also perform a hitting-only task without Go/No-go decision-making (NJ task) to compare to the hitting task with Go/No-go decision-making (J task). Using hand movement profiles, we evaluate decision-making and hitting movement. Our methodology uses movement profiles and so provide a powerful paradigm that can measure decision-making and movement simultaneously but allow for separate analysis. We hypothesize that the Go/No-go judgment is effective for increasing task performance, but the efficacy is reduced under short time constraints because of the trade-off between judgment and movement accuracy. However, movement strategy can be changed so as to compensate for the reduction in judgment efficacy. Our results demonstrate that Go/No-go decision-making enhances hitting task performance, but only if the time constraint is not <0.5 s. The drop in decision-making accuracy with tighter time constraints appears to deny enhancement in hitting task performance. However, deliberately slowing the onset of movement or canceling the movement in progress can ameliorate the drop in decision-making accuracy.

## Results

**Task performance and judgment accuracy according to TTCs.** Targets were released at six angles, $\pm 5°$, $\pm 8°$, and $\pm 15°$, from the frontal center in pseudo-random order so that the targets with $\pm 5$ and $\pm 8$ angles passed through the strike area (Strike target) while the others did not (Ball target) (Fig. 1b). The target arrival times from start to the strike area (time-to-contact: TTC) were 0.4, 0.5, and 0.6 s and implemented as separate sessions. Twelve participants performed two different tasks in separate sessions. In the Go/No-go judgment task (J task), participants were required to hit only the Strike targets (Go) in the strike area and to not hit the Ball targets (No-go). The other task (NJ task) required the participants to hit all targets with no judgment in the same timely manner as the J task. An example of hand trajectories is shown in Fig. 1b. Using hand movement profiles, we classified the observed movements into four possible judgment responses based on signal detection theory[29], i.e., Hit (Go in Strike), Miss (No-go in Strike), False Alarm (FA; Go in Ball), and Correct Rejection (CR; No-go in Ball) (see Supplementary Fig. 1a, b). As the judgment accuracy, we assessed sensitivity d′, a statistical measure to quantify how a system distinguishes signal distribution from noise distribution, for each TTC by calculating the ratio of four responses, Hit, Miss, FA, and CR. The Hit responses were further divided into two responses according to goal success, e.g., Hit-Goal and Hit-No Goal. We assessed goal rates (correct hitting; Hit-Goal rate to all Strike trials) and overall task success rate for J task, the J task success rate, which was defined as the ratio of the sum of the correct hitting (Hit-Goal) and the correct judgment (CR) responses in all trials (see Methods). Corresponding task success rates for NJ task, NJ task success rate, were also evaluated as the ratio of the correct hitting.

Both the J task success rate and the NJ task success rate decreased with shorter TTCs, but the trends differed between tasks (Fig. 2a). A two-way analysis of variance (2-way ANOVA) showed significant main effects of TTC ($F$ (2, 66) = 35.22, $p < 0.0001$) and task ($F$ (1, 66) = 29.90, $p < 0.0001$); there was no significant interaction ($F$ (2, 66) = 3.58, $p = 0.033$). The Post-hoc Bonferroni-corrected t-tests showed significant differences between tasks with 0.5 s ($p = 0.0078$) and 0.6 s TTC ($p = 0.00019$), but none at 0.4 s TTC ($p = 0.97$). With regard to the task J success rate, there were significant differences between 0.4 s and 0.6 s TTC ($p < 0.0001$), and between 0.4 s and 0.5 s TTC ($p < 0.0001$). As for the NJ task success rate, there were significant differences between 0.4 s and 0.6 s TTC ($p = 0.0038$). These results indicate that Go/No-go judgement is effective for task performance, since there were some differences in task difficulty between Strike and Ball targets

(Supplementary Fig. 2). However, importantly, this judgment enhancement was almost lost when TTC became <0.5 s. We found a significant reduction in the goal rate (correct hitting rate to Strike trials) for J task at 0.4 s TTC compared with the goal rate for J task ($p = 0.010$, Fig. 2b), although there was no significant main effect of task ($F$ (1, 66) = 2.06, $p = 0.156$) and significant interaction ($F$ (2, 66) = 5.49, $p = 0.0063$), indicating that making judgments under tight time constraints interfered with hitting accuracy. With regard to the goal rate for J task, the Post-hoc Bonferroni-corrected $t$-tests showed significant differences between 0.4 s and 0.6 s TTC ($p < 0.0001$), between 0.4 s and 0.5 s TTC ($p < 0.0001$), and between 0.5 s and 0.6 s TTC ($p = 0.038$). As for the goal rate for J task, there were significant differences between 0.4 s and 0.6 s TTC ($p = 0.023$). Figure 2c shows the mean judgement accuracy (i.e., d′) for each TTC. We found that judgment accuracy decreased significantly as TTC fell ($F$ (2, 33) = 39.77, $p < 0.0001$), with a pronounced reduction from 0.5 s to 0.4 s TTC ($p < 0.0001$). The reduction in the J task success rate was mainly caused by lower judgement accuracy, since the NJ task success barely changed from 0.5 to 0.4 TTC (Fig. 2a).

**Relationships between Go movement and judgement features.** Reducing TTC inevitably reduces the total time available for decision-making and movement execution. Though there are many studies about each SAT in decision[2–7] and in movement[8–10], little is known about how both SATs interact under severe time constraints. Thus, we evaluated some temporal features, judgment time, movement onset time, and movement duration, based on the hand acceleration profiles in the forward (y) direction in Hit trials (see Methods and Fig. 1c and Supplementary Fig. 3a–c). The judgment time was defined as the time at which the Hit trial diverged from the average No-go trial. The movement onset time was determined as the time when the Y direction acceleration exceeded 10% of its maximum value. The movement duration was defined as the time from onset time to target contact time. We found that movement onset time, judgement time, and movement duration significantly decreased, in similar proportions, as TTC decreased (Supplementary Fig. 4a, b) (Mean ± SD judgment time = 239 ± 31.8, 305 ± 40.5, 366 ± 42.4 ms, onset time = 226 ± 39.1, 283 ± 57.8, 335 ± 60.7 ms, movement duration time = 226 ± 39.1, 283 ± 57.8, 335 ± 60.7 ms, for 0.4, 0.5 and 0.6 s TTC, respectively), indicating that the time ratio of decision-making to movement execution was constant regardless of the time constraint (Supplementary Fig. 4c). Further we found that movement onset and judgement conclusion almost coincided, although there was a significant difference ($F$ (1, 66) = 4.05, $p = 0.048$) (Supplementary Fig. 4a). Indeed, we found a strong association between movement onset time and judgment time ($r^2 = 0.863$, $p < 0.0001$, Fig. 3a). In addition, movement onset time was significantly correlated with judgement accuracy ($r^2 = 0.372$, $p < 0.0001$, Fig. 3b). These results suggest that the movement strategy of delaying the start of movement raises decision accuracy, probably due to greater evidence accumulation with an increase in the continuous flow of evidence from the decision-making process into the motor preparation and motor execution processes[30–35].

**Changes in No-go movement pattern according to TTCs.** The drop in judgment accuracy due to shorter TTC included decreases in the rate of Hit responses (HR) and in the rate of CR responses (CRR) (Supplementary Fig. 5). In particular, the CRR decrease tended to be higher than that of HR, indicating that No-go decision was more hindered by severe time constraints than Go decision. In association with this decisional difference, we found that participants were apt to move their hand even in No-go trials as TTC fell ($F$ (2, 33) = 3.35, $p = 0.047$, Fig. 4a, b). We hypothesize that this is due to insufficient time for Go/No-go

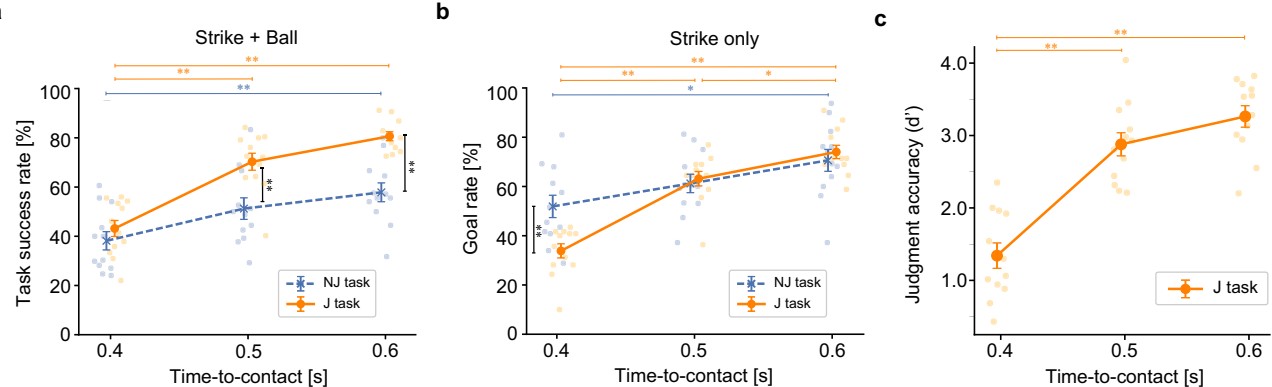

**Fig. 2 TTC-dependent changes in task success rate, goal rate, and judgment accuracy. a**, **b** The mean task success rate for Strike and Ball targets (**a**) and the mean goal rate for only Strike targets (**b**) for each TTC are plotted as orange lines (the J task success rate) and blue dashed lines (the NJ task success rate) (Mean ± SEM, $n = 12$, *$p < 0.05$, **$p < 0.01$). The individual data points are overlaid. **c** The mean judgement accuracy ($d'$) for each TTC is shown by orange lines (Mean ± SEM, $n = 12$, *$p < 0.05$, **$p < 0.01$) with individual data points overlaid. Please note that the J task success rate and judgment accuracy decreased as TTCs were shortened, and judgment was ineffective in terms of J task success rate if TTC was <0.5 s.

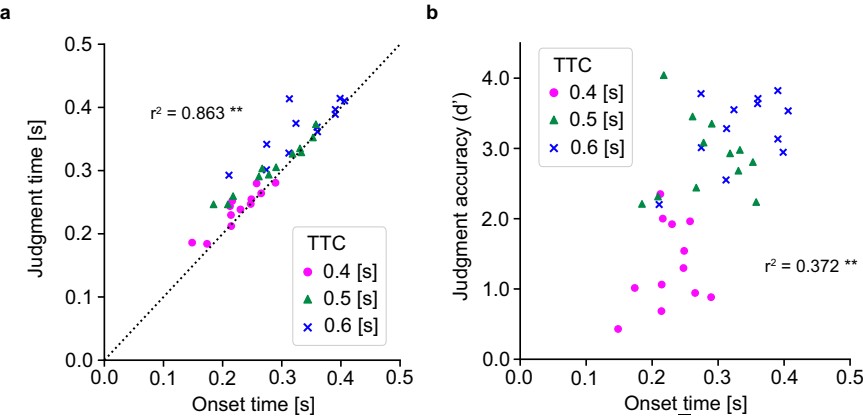

**Fig. 3 TTC-dependent relationship between judgment and movement features. a**, **b** Relations between the mean movement onset time and the judgement time (**a**) and the judgement accuracy ($d'$) (**b**) for each TTC (0.4 s: magenta circle, 0.5 s: green triangle, and 0.6 s: blue cross). $r^2$ represents the coefficient of determination. ($n = 12$, **$p < 0.01$). Please note that the movement onset time was strongly correlated with judgment time and judgement accuracy.

judgement and hitting movement. Accordingly, we estimated the time obtained by adding the judgement time at each TTC (Supplementary Fig. 4a) to the movement duration at 0.4 s TTC as the minimum demanded movement duration (Supplementary Fig. 4b), which means the time necessary for making judgement and movement execution at each TTC. The results show that the desired time at 0.6 TTC was below the actual TTC, while that at 0.4 TTC exceeded the actual one (Supplementary Fig. 6), indicating that if TTC is less than 0.5 seconds, Go/No-go judgement must partially overlap movement execution.

Participants, therefore, would change their movement strategy so that they moved their hand once and then stopped it when the judgement was No-Go. Importantly, the success of this stopping movement (move-then-stop) partly enhanced judgement accuracy. We found a significant correlation between the success rate of move-then-stop movements and judgement accuracy ($d'$) at 0.4 TTC ($r^2 = 0.69$, $p = 0.0083$, Fig. 4c), suggesting that this move-then-stop strategy can suppress the drop in judgement performance imposed by strong time constraints.

## Discussion

The theoretical view of how we make sensorimotor decision assumes mutual interaction between decision-making and action.

Well known as the embodied decision theory, it involves the contributions of action and motor processes to decision-making and vice versa[11–13]. Our findings correspond with this theory reasonably well. First, we observed the fall in task performance when TTCs decreased (Fig. 2a, b). It is conceivable that this is basically caused by the shortening of judgement time and movement duration (Supplementary Fig. 4a, b) which induces drops in judgement and movement accuracies, respectively (Fig. 2a, c). Though each assessment variable was evaluated in every session (including the maximum number of 120 trials), we observed no clear temporal evolution within session for the task success rate, the decision accuracy and changes in movement strategic variables, such as delaying the movement onset or move-then-stop, except the goal rate (Supplementary Fig. 7). This indicates that learning within session occurred infrequently. The results of the inter-subject variability analysis between goal rate and the movement strategies (move-then-stop and movement onset) revealed that the trial-to-trial changes in movement strategies had virtually no impact on the improvement in the goal rate (Supplementary Fig. 8).

As shown by Battaglia and Schrater[16], in a situation where the participant had enough time to judge and then move, the time allocation of perceptual decisions and movement execution could be optimally distributed. Choi et al.[17] have also demonstrated the

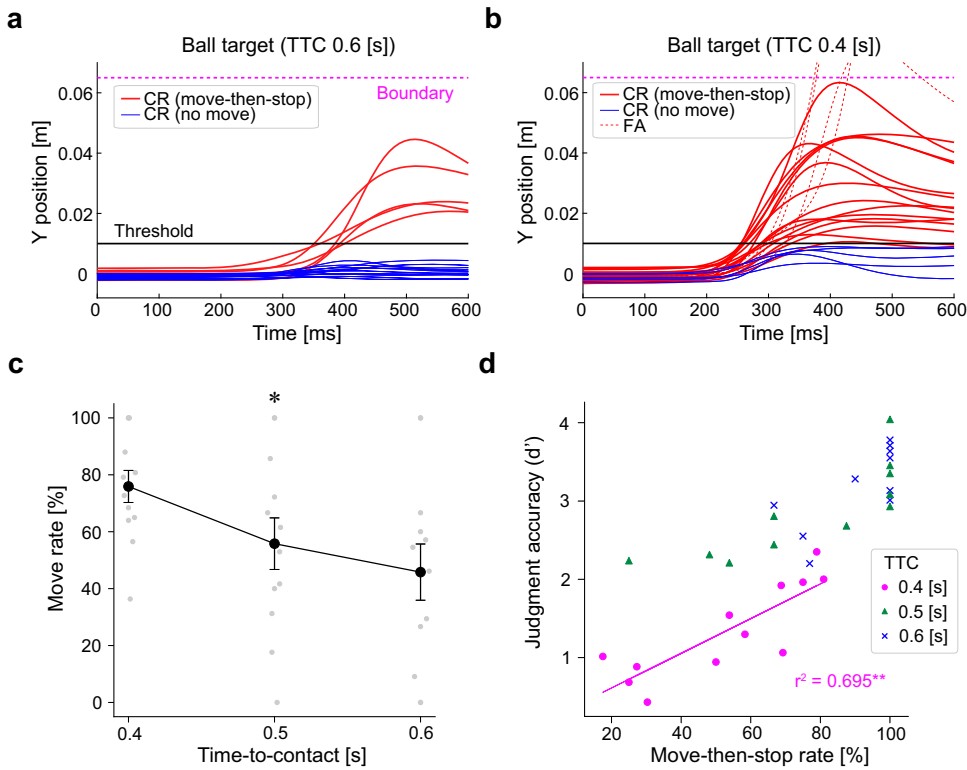

**Fig. 4 TTC-dependent changes in No-go behavior and their impact on judgment accuracy. a, b** Example of Y position profiles of Ball targets in J task in a representative participant. The Y position profiles for CR (move-then-stop) trials, CR (no move) trials, and FA trials are plotted as red lines, blue lines, and red dashed lines, respectively, for 0.6 s (**a**) and 0.4 s TTC (**b**). The magenta dashed line indicates the Go/No-go boundary and the black line indicates the threshold that determined whether the cursor moved or not. **c** The mean move rate of CR trials for each TTC in J task (Mean ± SEM, $n = 12$) shown by black lines with individual data points overlaid. There is a significant main effect for TTC (*$p < 0.05$). Please note that due to the decrease in TTCs, participants did not hold the hand stationary, but moved their hand once and then stopped it when No-go judgement was required. **d** Relation between the success rate of move-then-stop trials and judgement accuracy (d') for each TTC in individual participant (0.4 s: magenta circle, 0.5 s: green triangle and 0.6 s: blue cross). The magenta line shows the regression line in 0.4 s TTC plots and $r^2$ represents the coefficient of determination. ($n = 12$, **$p < 0.01$). Please note that the move-then-stop strategy is effective in suppressing No-go judgement mistake under severe time constraint.

cost of sharing time between decision-making and motor control for periods of seconds. Other studies have argued that urgent signals, assumed to be temporally strengthening signals related to the urgency in making a decision[36], increased the speed of triggering reaching movement and the vigor of saccades[17,37], suggesting a shared mechanism between decision-making and motor processes that involve the basal ganglia[17,37,38]. A recent study on manipulating motor contexts showed that slow movements executed for demanding targets were often preceded by faster and less accurate decisions, indicating that participants sacrificed decision-making for action execution[19]. This suggests that decision urgency and movement vigor signals are likely to interact. Congruent with these studies, our results indicate that the time allocation between decision-making and motor execution is kept even for durations under one second (Supplementary Fig. 4c). However, the relative increase in time cost penalty with severe time constraints is likely to erase the efficacy of decision-making (Fig. 2c) so that the equilibrium between the two tradeoffs (speed (time) vs. accuracy) of decision-making and movement execution, is disrupted.

In Go trials, the movement onset time was strongly correlated with the judgment time; they occurred at almost the same time (Fig. 3a). This resulted in the correlation between movement onset and judgment accuracy (Fig. 3b), suggesting that the movement strategy of delaying the start of the movement would be effective. We also found that time proportion of movement onset and decision to TTC were constant among the TTC conditions

examined (Supplementary Fig. 4c). A possible interpretation is that the movement onset as well as decision timing would be comprehensively processed according to the time constraint. These movement and decision could be made based on sensory information at a certain time (i.e., target position) or on urgency signal-related accumulated evidence (i.e., target velocity and motion) in combination with other information such as strike area and hand cursor. Participants are likely to utilize such task-relevant information to judge whether the target will pass through the strike area (Strike) or not (Ball) and to estimate the position to be reached. Many studies have shown that the decision process and motor planning process, such as target selection and action specification[39], run in parallel[40–43]. Our finding of a correlation between movement onset time and judgement time is consistent with these studies, since movement onset (execution) generally follows movement planning. On the other hand, other studies have demonstrated the impact of motor control, i.e., motor cost[44–46], on decision-making (see reviews[14,15]). Lepora and Pezzulo[12] proposed the idea that a feedback loop from the action to the decision-making process produces commitment where the action completion triggers decision completion. Our result suggests that movement and decision processes occurred concurrently. Our results provide another piece of evidence for motor control being linked to decision-making.

We also found that as TTC fell, CRR decreased more than HR (Supplementary Fig. 5). Previous studies have reported a similar relationship between CRR and HR[47], and a difference in activation areas as captured by functional MRI between Go and

No-Go responses[48]. These facts imply that No-Go decision and Go decision are not just two sides of the same coin, and No-go decision is more difficult than Go decision if the time constraint is severe. Interestingly, we further observed that participants moved their hand once and then stopped in No-Go trials, and the success of such movement (move-then-stop) was correlated with judgement accuracy (Fig. 3c), suggesting that the ongoing movement strategy changes with the level of time constraints. This is reasonable because it takes more time to achieve judgement and movement in a serial manner, that is, completing judgement before movement execution. If participants select Go movement in advance and make Go/No-go judgement in parallel with the ongoing movement, they can secure more time in which to improve judgement accuracy. Shorter visuomotor delay in continuous movement compared to discrete movement[49] can be also related with change to ongoing go-stop strategy. Actually, in the task known as the go-before-you-know task, which required the participants to initiate a reaching movement towards potential reach targets before knowing the final target location (target cue was shown after movement onset), it has been shown that participants launched reaching movements to a position intermediate to potential targets[50]. This spatial averaging strategy reflects optimizations based on task constraints and motor costs. However, when reducing the time available for corrective movements, the movement strategy changed to 'go to one of the targets' rather than spatial averaging, indicating changes in movement strategy according to time constraints[51]. Embodied decisions are made by competition between internal representations of potential action[34,52,53], which can be regarded as affordances defined as the perception of possibilities for, or restraints on, an action that the environment offers[54,55]. These affordance competition hypotheses proposed that these affordances could be biased by the desirability of the expected outcome or the necessary effort[46,56,57]. In light of the affordance competition hypothesis, our task could provide participants with an available affordance (a reachable strike area by moving the hand) and another affordance (a start position without moving the hand). The former could produce the expected affordances that those actions make available (i.e., hitting the target and returning it to the goal or resulting in FA), and the latter could produce the expected affordances (no hitting in CR). These expected affordances would impose a top-down bias in favor of hitting the target (a reachable strike area by moving the hand) when the time comes too short for accomplishing the task by initiating movement after the decision, which could cause change in movement strategy that the move rate increased as the time constraints became shorter. This is because task completion was to return the target to the goal area 20 times. We speculate that affordances could be biased with time constraints by urgency signals[36].

Since our task involved motor inhibition, it triggered not only activation (Go) but also the inhibition (stop) strategy. The inhibitory process usually takes place ~100–200 ms after the onset of the stop stimulus[58–60]. Current models of inhibitory control propose that the processes of inducing (Go) and withdrawing (Stop) the behavior are nearly independent, ultimately interacting to inhibit potential or actual behavioral responses when a stop signal is externally presented[61–63]. Some neural mechanisms in the brain would be involved with such inhibited action. It was recently shown that the pre-supplementary motor area as well as the supplementary motor area are involved in response inhibition[64,65]. Other studies have shown that other brain areas, such as, inferior frontal gyrus, and the subthalamic nucleus, the basal ganglia, play important roles in suppressing unwanted movements[66–68]. Adjustment of the movement

strategy, especially action inhibition, would be probably regulated by the basal ganglia[67,68] and related areas.

Our research has some limitations that should be addressed. Many studies have shown common or coordinated control of eye and hand movements in interceptive tasks, but we did not measure eye movements in this study. Fooken and Spering[69] showed that smooth pursuit and saccadic eye movement parameters provide reliable estimates of Go/No-go manual intercept. This indicates that eye movement modulations reflect an early readout of decision formation itself. Owens et al.[22] also designed a baseball-specific Go/No-go task with screen touch under a time constraint of 0.4 s to 0.6 s comparable to our study, and showed that baseball experts showed better gaze control than novices at the short preparation and fast stimulus speed trials (0.4 s TTC) including significant differences in stimulus tracking, gaze to stimulus distance, and peak pupil dilation. Whereas saccade onset, hand movements, and screen touch reaction times did not differ between groups. Additional research is needed to clarify how eye-movement kinematics relate to decision making and motor execution under the time constraint. There are also some gaps between our experiment and actual sports scenes. For example, though each TTC corresponding to target speed was fixed for each block, in actual games the value would be more random. In addition, other factors, such as individuality (like sports experience) and ball type prediction from the pitching form, could strongly impact decision accuracy. Despite these limitations, this research offers future growth potential from our elucidation of the relationship between movement and judgment under severe time constraints; it can differ drastically from what we would normally expect.

In conclusion, we have demonstrated the potential limits of balancing rapid decision-making and motor control and the corresponding change in ongoing movement strategy to compensate the attenuation of judgement accuracy imposed by severe time constraints. This is best demonstrated by the case of baseball where judging a strike or ball is useless with pitch speeds of 100 mph (corresponds to 0.4 s TTC) or more. However, the ability to adjust swing behavior, that is, slowing the swing onset and stopping a swing, are key skills for improving hitting success.

## Methods

**Participants**. Twelve volunteers (eight men and four women all right-handed, aged 24–41 years: Mean ± SD age, 32.58 ± 4.68 years) participated in the experiment. None had motor or visual disorders and all had normal or corrected-to-normal vision. The sample sizes were determined based on the minimum demand for realizing counterbalanced conditions.

**Ethics statement**. All subjects gave written informed consent prior to participation. All experimental protocols were approved by the Ethics Committee of NTT Communication Science Laboratories (H26-007) and were in accordance with the Declaration of Helsinki.

**Experimental setup**. Participants were seated in front of a manipulandum (KINARM End-Point Lab, BKIN Technologies, Canada), looked down at a horizontal mirror reflecting the display of a 47-inch LCD monitor (47LS35A, LG), and grasped the handle below the mirror with their dominant hand (Fig. 1a). Participants did not see their arm, and the cursor (orange circle with a radius of 1 cm) on the mirror indicated the position of their hand. The cursor start position (0 cm, 0 cm) was shown as a red circle with radius of 1 cm and the center of the circle was defined as the origin. The goal area to which the target was to be returned was shown as a 60 cm × 3 cm pink rectangle at the center position (0 cm, 41 cm). A 12 cm × 3 cm green rectangle placed horizontally indicated the strike area and the center position (0 cm, 11 cm). The width of the goal area was designed so that the angle between lines from each edge of the goal area to the center of the strike area was 90°, imitating a baseball diamond. The target was shown as a white circle with a radius of 1 cm, and was displayed at the same position (0 cm, 41 cm) as the center of the goal area at the trial start. These stimuli were provided using MATLAB, Simulink and Stateflow (MathWorks). The drawing refresh rate was 60 Hz, and the

delay was up to 50 ms. All experimental data were collected at a sampling rate of 1 kHz through the manipulandum.

**Procedure and conditions**. The experiment was a two-factor in-participant experiment with three levels of TTC (0.4, 0.5, and 0.6 s) and two levels of task (J task and NJ task). When the cursor stayed at the cursor start position for 200 ms, a beep sounded and the target started to move downward on the Y-axis. There were six angles at which the targets were released: ±5°, ±8°, and ±15° with respect to the downward direction of the Y-axis (Fig. 1b). Targets traveling at ±5° and ±8° were Strike targets, while targets with ±15° were Ball targets. The angle of the target was pseudo-randomly intermixed with the same probability, meaning the probability of Ball trial was ~33.3%. The target moved at constant velocity, 0.75, 0.6, and 0.5 m s⁻¹ yielding TTC values of 0.4, 0.5, and 0.6 s, respectively. There was a margin in the time the target took to pass through the strike zone, 400 ± 33 ms, 500 ± 41 ms, and 600 ± 50 ms for 0.4, 0.5, and 0.6 s TTC, respectively. The direction and velocity of the returned target were calculated assuming an inelastic collision with a collision coefficient of 0.5, the cursor and the target are circles of the same mass.

Participants were required to hit the moving target in a timely manner as above so as to return it to the goal area. In J task, participants were required to hit only Strike targets and not to hit Ball targets. Countermanding of the movement was allowed even if the participant had started hand movement as long as cursor did not exceed the boundary before the target passed by (Supplementary Fig. 1a). The boundary line was defined as the forward ($y$) position one target diameter below the bottom line of the strike area ($y_{(center\ of\ the\ cursor)} = 6.5$ cm). When participants hit the target within the strike area, the returned target turned yellow and rebounded. If participants hit the target outside the strike area or hit a Ball target, the target turned red and stopped immediately. In NJ task, participants were required to hit all targets regardless of Strike or Ball. In both tasks, the Strike target was required to be hit within the strike area. In NJ task, Ball targets were allowed to be hit outside the strike zone with the same timing as J task (the image of the strike zone was expanded horizontally). Participants first performed one session consisting of each TTC in one task, and then performed another session consisting of each TTC in the other task. Each TTC was implemented as an independent block. The order of the tasks and the order of the TTCs within the task were counterbalanced between participants. There were six blocks in all, and the task goal in each block was to hit the target into the goal area twenty times. During the experiment, in J task, trials in which participants could return the target to the goal area only for Strike targets were counted as successful goals, and in NJ task, all trials in which participants could hit the target into the goal area were counted as successful goals. Trials in which returned target speed was slow and it took more than one second to reach the goal area from contact were classified as Hit-No Goal trials. The maximum number of trials in one block was set to 120 (up to 20 for each angle), and even if the task goal was not achieved, the block was terminated when the number of trials reached 120. Progress within each block was indicated by the height of the green bar on the right side of display so that participants could check it.

**Statistics and reproducibility**

*Exclusion criteria*. For two of the twelve participants, data of up to 120 trials were analyzed because they could not achieve the task goal with 0.4 s TTC. Trials where the hand was moving at the target start (acceleration exceeded 0.4 m s⁻²) were excluded as error trials. Error trials were 1.4% of all trials. The average number of trials with 0.4 s, 0.5 s, and 0.6 s TTC were 88.50 ± 20.26, 48.67 ± 11.16, and 40.42 ± 5.26, respectively for J task, and for NJ task they were, 51.17 ± 17.98, 41.75 ± 11.83, and 36.42 ± 10.06 (Mean ± SD), respectively.

*Data analysis*. The data was smoothed using a double pass filter with a cutoff frequency of 10 Hz. It is known that the hidden decision-making process can be clarified by examining the hand trajectory toward the target, for example, from the degree of curvature of the trajectory[34]. Therefore, we decided to elucidate the decision-making process based on the position of the hand, that is, to classify the reactions. The following judgment and motion criteria were set in the y direction. This is because when considering actual baseball batting scenes, the lateral direction to the pitcher is not important for judging whether the batter has swung, only movement in the front-back direction is important.

The criteria for determining whether the Go/No-go judgment was correct was defined as whether the cursor moved over the boundary (Supplementary Fig. 1a). In J task, if the cursor passed the boundary, the trial was defined as a Hit response for Strike targets or defined as a False Alarm response (FA) for Ball targets. Conversely, if the cursor did not cross the boundary, the trial was defined as a Miss response for Strike targets or defined as a Correct Rejection response (CR) for Ball targets. We calculated the rates of each result, Hit rate (HR), Miss rate (MR), False Alarm rate (FAR), and Correct Rejection rate (CRR), and calculated sensitivity d′ from the Z-score of FAR and HR by the signal detection theory[29]. In calculating each rate, because HR = 1 or CR = 1 might occur when TTC was longer than 0.5 s, each rate was adjusted based on the number of trials[70]. We defined sensitivity d′ as the judgment accuracy. HR and CRR are shown in Supplementary Fig. 5.

We defined the J task success rate and the NJ task success rate, which regarded as total task performance, and the goal rate for Strike targets, as follows (see also Supplementary Fig. 1b).

$$\text{J task success rate (Strike + Ball)} = \frac{\text{Hit-Goal} + \text{CR}}{\text{Total}} \quad (1)$$

$$\text{NJ task success rate (Strike + Ball)} = \frac{\text{Goal}}{\text{Total}} \quad (2)$$

$$\text{goal rate (Strike only)} = \frac{\text{Strike-Goal}}{\text{Strike}} \quad (3)$$

The J task success rate was defined as the number of Hit-Goal and CR responses divided by the total number of trials, and the NJ task success rate was defined as the number of goal responses divided by the total number of trials. In order to examine the impact of judgment on hitting accuracy, we determined the goal rate (Strike). It was defined as the number of Strike-Goal responses divided by the number of Strike trials for J task and NJ task.

Using the Y direction acceleration data in J task, the judgment time for Hit responses was determined as the time when the value obtained by subtracting the average data of all No-go (CR and Miss) trials from each Hit response data exceeded the last time the threshold value was exceeded (Fig. 1c and Supplementary Fig. 3a–c). The threshold for each trial was set as 10% of the maximum value of the above subtraction. The reason for combining the average data of Miss and CR responses as all No-go trials was that there was no FA or Miss response for some participants, especially for long TTCs. The movement onset time was determined as the time when the Y direction acceleration data exceeded the threshold value, and the threshold for each trial was set as 10% of its maximum value. The movement duration was defined as the time from onset time to target contact time in Hit response.

In order to verify how the movement behavior changed in the No-go trial, the following two rates were calculated. The move rate for Ball targets was defined as the rate of trials in which cursor movement exceeded 1 cm (the radius of the cursor) from the start point. The move-then-stop rate for Ball targets was defined as the rate of successful countermanding trials, which participants started to move but stopped correctly (=CR). And then, in order to examine what caused the behavioral change with the decrease in TTC, we estimated the time required when judgment and movement were assumed to be sequential by adding the movement duration at 0.4 s TTC of NJ task (Supplementary Fig. 4b), which was the shortest movement duration, to the judgment time at each TTC (Supplementary Fig. 4a). The estimated time was compared with the actual time, at each TTC (Supplementary Fig. 6).

**Statistics and reproducibility**. We tested for differences in all outcome parameters between tasks and TTCs using two-way analysis of variance (two-way ANOVA) tests. The Post-hoc Bonferroni-corrected tests were used to examine the differences between TTCs and tasks. For testing the difference in all outcome parameters in J task due to the change in TTC, we used one-way analysis of variance (one-way ANOVA) tests. To calculate the correlation, Pearson's correlation coefficient was used. Statistical significance was set at $p < 0.05$ for all tests. The sample size was 12. All statistical testing was performed using Python (v3.7.1, Python Software Foundation, Delaware, America).

**Reporting summary**. Further information on research design is available in the Nature Research Reporting Summary linked to this article.

## Data availability

The datasets generated during and/or analyzed during the current study are available from the corresponding author on reasonable request. Source data for figures in the paper is available in FigShare: https://doi.org/10.6084/m9.figshare.19106813.v2. All other data are available from the corresponding author on reasonable request.

## Code availability

For data analysis, we used MATLAB (version R2018b) with the Signal Processing Toolbox and Python (version 3.7.1). The computer code is available upon reasonable request from the corresponding authors.

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

## Acknowledgements

We thank Sho Ito for excellent technical support (KINARM), and Makio Kashino and members of the Kashino Diverse Brain Research Laboratory for useful discussions.

## Author contributions

Conceptualization by A.K. and T.K.; Data curation by A.K.; Formal Analysis by A.K.; Methodology by A.K. and T.K.; Investigation by A.K. and T.K.; Software by A.K.; Validation by A.K. and T.K.; Project Administration by T. K.; Writing – Original Draft by A.K. and T.K.; Writing – Review & Editing by A.K. and T.K.; Funding acquisition by T.K.; Supervision by T.K.

## Competing interests

A.K. and T.K. are affiliated with Nippon Telegraph and Telephone Co. The authors have no other competing interests to declare.
