## [Peer Review File · Communications Biology]

Reviewers' comments:

Reviewer #1 (Remarks to the Author):

Article Title: Compensative movement ameliorates reduced efficacy of rapidly embodied decisions

The authors aimed to investigate how short time constraints affect decision making and movement accuracy related to hitting a ball during a baseball-like task. They set up two types of task: a hitting only task and a hitting with decision making. They enrolled participants. They showed that Go/No-go decision making enhances hitting task performance, especially for time constraints that are superior or equal to 0.5 seconds. Decision-making accuracy with tighter time constraints worsen hitting task performance. However, strategic behaviors emerged under tighter time constraints: slowing the onset of movement or canceling the movement in progress could improve decision making accuracy for the Go/No-go task compared to hitting only task. These results have implication on theoretical neuroscience and on strategies for sport training.

The novelty of this study is that the two tasks allowed the authors to explore decision making process and the movement kinematic separately or conjointly using hand movement profiles. The study is clearly written and nice to read.

Here are the comments that the authors should address to clarify or deepen some points.

Major comments

1. A priori hypothesis. There is no clear definition of the tested hypothesis in the introduction. If the study was exploratory, the authors should make a clear mention of that fact, or introduce earlier studies. For instance, it seems that the concept of visuomotor delay could play a role in the present study. Visuomotor delay is The duration of visuomotor delays appears to depend on the subsequent manner in which the information is used. It can be as short as 100 ms when information is used during the on-line regulation of the movement (Bootsma and van Wieringen, 1990; Savelsbergh et al., 1991), but can reach values near 200 ms when information is used to produce movement relying on decision processes or more discrete movements, such as the beginning of the movement or some important correction of that movement (Lee et al., 1983 ; Michaels et al., 2001 ; Benguigui et al., 2003). A study evaluated the effect of shorter latencies of time-to-contact on hand movement profiles when hitting balls in a group of non-expert as well as expert tennis players (Le Runigo, Bardy and Benguigui, 2005 Human Movement Science). Participants had to adapt their movements by making decisions (increase or decrease hand movement velocities) at 0.4 s and 0.26 s before time to contact, in delays compatible with the present study. This could represent a rationale to build hypotheses about embodied decision and movement kinematics under tight timing constraints.

If needed, these references are in:

2. Eye-movement kinematics. Many decisions such as to swing at a baseball pitch are accompanied by characteristic eye-movement behavior, preceding the onset of hand movements. Using a Go/NoGo paradigm, Go/NoGo decisions were reflected in distinct eye-movement responses (Fookien and Spering, 2019; Journal of Vision). Under tight time constraints between the moment when the ball appears in the visual field of view and the moment when one has to initiate a hand movement or not, decision making relies on the speed with which visual information can be detected, and might depend on eye-hand coordination. Did the authors measure eye-movement kinematics? Could eye-movement kinematics help to refine the link between decision signals and hand motor actions?

3. Learning processes and inter-subject variability. Participants performed consecutive trials for each Time To Contact (TTC) condition, separated in 6 blocks. There was a maximum of 120 trials within each bloc. This gives the possibility to look at learning profiles (temporal evolution of judgement accuracy and/or task success rate). Does inter-subject variability in learning profiles are related to specific strategies of delaying movement onsets/cancelling the ongoing movement, especially under tight temporal constraints? How early in the learning process individuals that have better success rates are able optimize their behavior?

4. Task-relevant resources. Judgement time is driven by the information that participants could

process to initiate their behavior. In the present study, what information is considered to initiate these processes? It might be important to identify the task-relevant information and resources the agent has access to in order to solve the task.

5. Embodied cognition/NoGo judgements under severe timing constraints. The results of No-Go judgement are very interesting and should be put forward for its full potential, i.e. for its implication on our understanding of the behavior. Tasks and skills can be trained to improve behavior. This can be underlined by mental representations, but there is a point when the mental representations are not as useful for skill development. In case of tight timing constraints, the skill becomes more instinctual and based on the experience of trial-to-trial feedback that update what information to rely on to efficiently link perception and action. With practice, participants are able to make judgements and decisions for the best possible outcome more quickly and efficiently based upon the affordances available but without having to take the time to consider all possible outcomes or consequences of each decision. As the system evolves and grows, the outcome of these judgements and decisions add upon themselves and create a new set of initial conditions that lead to new judgement and decision-making behaviors. My question is linked to the previous comment (#3). What are the conditions in the present task that make the participant use the strategy go-stop? At which point along motor repetitions the consequences of one decision feedback onto itself to create an optimal set of judgements that lead to accurate decisions? Is this point occurring earlier or later during tight compared to larger timing constraints?

Minor comments

1. Introduction: lines 62-63, however is repeated twice. Double opposition makes the reasoning difficult to follow.

2. Population: Since expertise plays a role in decision making regarding rapid movements, how expert were the participants in regards to baseball (occasional players or naïve versus regular players)? Were the participants paid for their participation?

3. Results. The variables that were submitted to the statistical analyses were not all introduced. For instance, results of judgement time are presented in the second section of the Results. While the reader can guess what movement time is, judgement time deserves a quick explanation without having to dig into the supplemental material, for a matter of fluidity in the reading process. Maybe a figure of hand movement profile would help to dissociate or make the link between the different variables, including the judgement time.

4. Discussion lines-199-200 . "Our result suggests that movement onset might force decision completion." This statement is difficult to understand when we consider that for the 0.4 s condition, movements were initiated and stopped, which was correlated with judgement success. This rather suggests that the relationship between movement onset and the decision completion is more complex and depends on the timing constraints: Movement onset can also occur independently from the final decision to move or not to move in situation when timing constraints are severe.

Reviewer #2 (Remarks to the Author):

Review of the article "Compensative movement ameliorates reduced efficacy of rapidly embodied decisions"

The study focuses on the influence of available time in a task where decision time and movement execution are sequentially performed, and the overall time to assess and execute the movement is constrained. The task consists of hitting (Go) (or not; NoGO) a ball within a given time interval, presented along one of six inclinations.

Main Comment

My impressions is that this paper deals with a very interesting topic, which is the relationship between decision-making between actions and motor control. Furthermore, under the interesting constraint of severe time pressure. I also believe the paper provides interesting results, in so far 0.5s seems to be the boundary to trade-off between decision-making and motor control in their current framework.

However, the general impression is that the paper requires some additional general work. It lacks structure, the proper introduction of hypotheses, the experiments are not properly introduced. It is in general difficult to read and therefore hard to understand. I cite a few of the problems next:

- The notion of TTC is crucial in the experiment, and is not properly described until very late in the paper.
- Line 87. What is start area? It is nowhere to be seen in FIG 1.
- Lines 89-90. The description of the Strike vs Ball targets is first demands a little more introduction. Presentation of both tasks is difficult to understand because it is not properly introduced. Why the two tasks, and why in this specific fashion? In fact, I miss the clear formulation of a hypothesis that justifies and guides this particular experimental arrangement.
- Line 96. What is d' ? as a judgement accuracy?
- From Suppl FIG 1B it looks as if all No-Go targets are in the plus-minus 15 degree orientation. That would make the predictable. Does this carry any implication for their conclusions?
- Supl. FIG 1C-D, align these curves on movement onset, and make that the origin of time. They're hard to read just like that.
- The statistics must provide accurate p-values. Providing boundaries ($p < 0.01$ or $p < 0.05$) is not good enough.
- There is a number of papers providing some interesting insight onto the SAT they mention. The authors should consider basing their argument on this previous work. From the top of my head Reynaud et al. 2020. They already start dealing with time constraints, as does as well the Thura et al. 2014 you cite.
- In the discussion, the authors mention that movement onset might force decision completion. The view that a decision can be finished before the movement has been completed cannot be qualified as embodied.
- The definition of success rate across the three task is not homogeneous, as you are quantifying different metrics across different experiments. I would be cautious to call all of this success rate on equal terms.

Rebuttal Letter

Nov 22th, 2021

Responses to Reviewer #1

Thank you for the time and effort expended in generating the thoughtful and constructive feedback on our manuscript. Our responses to your comments are as follows. Corrected and added parts are noted in red in the manuscript and the Supplementary information.

Sincerely yours,

Akemi Kobayashi

2. Major comment 1. A priori hypothesis. There is no clear definition of the tested hypothesis in the introduction. If the study was exploratory, the authors should make a clear mention of that fact, or introduce earlier studies. For instance, it seems that the concept of visuomotor delay could play a role in the present study. Visuomotor delay is The duration of visuomotor delays appears to depend on the subsequent manner in which the information is used. It can be as short as 100 ms when information is used during the on-line regulation of the movement (Bootsma and van Wieringen, 1990; Savelsbergh et al., 1991), but can reach values near 200 ms when information is used to produce movement relying on decision processes or more discrete movements, such as the beginning of the movement or some important correction of that movement (Lee et al., 1983 ; Michaels et al., 2001 ; Benguigui et al., 2003). A study evaluated the effect of shorter latencies of time-to-contact on hand movement profiles when hitting balls in a group of non-expert as well as expert tennis players (Le Runigo, Bardy and Benguigui, 2005 Human Movement Science). Participants had to adapt their movements by making decisions (increase or decrease hand movement velocities) at 0.4 s and 0.26 s before time to contact, in delays compatible with the present study. This could represent a rationale to build hypotheses about embodied decision and movement kinematics under tight timing constraints.

If needed, these references are in:

- **RESPONSE:** As the reviewer pointed points out, visuomotor delay appears to be a reasonable basis on which to build our hypotheses. The Introduction (p.3, lines 62-69, 75-81) and Discussion (p.8, lines 266-267) now reference previous studies

on decision-making and visuomotor delay to explain the relationship between decision-making and motor process in terms of time constraints as follows.

- *“Severe time constraints are expected to impact both decision-making and motor execution. Regarding decision-making, Owens et al.²² assessed a baseball batting-specific Go/No-go decision responses, in which participants were required to reach the screen if the moving target passed within a certain area and not to reach if it passed outside the area. In this task, the time from target start to reach contact (Time-to-contact; TTC) was restricted to 0.4s, 0.5s, or 0.6s. Result showed that the decision accuracy decreased significantly when TTC reached 0.4s, but made no mention of the motor execution aspects such as movement duration and trajectory. Movement strategy can be also changed with time constraints.”* in the Introduction (p.3, lines 62-69).
- *“Visuomotor delay, which is defined as the time between the emergence of visually detectable information and the initiation of resulting motor adjustment²⁵, is a possible determinant of sensorimotor decision behavior. The extent of this delay has been shown to vary between 100ms and 300ms depending on the ongoing task, such as the online regulation of the movement²⁶ and the initiation of movement²⁷, and motor expertise²⁸. While these studies indicate that short time constraints affect decision-making and motor execution, it is unclear how they are balanced under severe time constraints.”* in the Introduction (p.3, lines 75-81).
- *“Shorter visuomotor delay in continuous movement compared to discrete movement⁴⁹ can be also related with change to ongoing go-stop strategy.”* in the Discussion (p.8, lines 266-267).
- We now also describe our hypothesis briefly in the last paragraph of the Introduction (p.4, lines 95-98) as follows.
 - *“We hypothesize that the Go/No-go judgment is effective on task performance, but the efficacy is reduced in short time constraints because of a trade-off between judgment and movement accuracies. However, movement strategy is changed so as to compensate for the reduction in judgment efficacy.”*

3. Major comment 2. Eye-movement kinematics. Many decisions such as to swing at a baseball pitch are accompanied by characteristic eye-movement behavior, preceding the onset of hand movements. Using a Go/NoGo paradigm, Go/NoGo decisions were reflected in distinct eye-movement responses (Fooker and Spering, 2019; Journal of Vision). Under tight time constraints between the moment when the ball appears in the visual field of view and the moment when one has to initiate a hand movement or not, decision making relies on the speed with which visual information can be detected, and might depend on eye-hand coordination. Did the authors measure eye-movement kinematics? Could eye-movement kinematics help to refine the link between decision signals and hand motor actions?

- **RESPONSE:** We didn't measure eye-movement kinematics because of limitations in the experimental design and measurements. However, as you mentioned, eye-movement kinematics and eye-hand coordination could refine the link between decision signals and hand motor actions. We discussed the role of eye-movement behavior in interceptive tasks in the Discussion (p.10, lines 303-315) as follows.

➤ *“Our research has some limitations that should be addressed. Many studies have shown common or coordinated control of eye and hand movements in interceptive tasks, but we did not measure eye movements in this study. Fooker and Spering⁶⁹ showed that smooth pursuit and saccadic eye movement parameters provide reliable estimates of Go/No-go manual intercept. This indicates that eye movement modulations reflect an early readout of decision formation itself. Owens et al²² also designed a baseball-specific Go/No-go task with screen touch under a time constraint of 0.4s to 0.6s comparable to our study, and showed that baseball experts showed better gaze control than novices at the short preparation and fast stimulus speed trials (0.4s TTC) including significant differences in stimulus tracking, gaze to stimulus distance, and peak pupil dilation. Whereas saccade onset, hand movements, and screen touch reaction times did not differ between groups. Additional research is needed to clarify how eye-movement kinematics relate to decision making and motor execution under the time constraint.”*

4. Major comment 3. Learning processes and inter-subject variability. Participants performed consecutive trials for each Time To Contact (TTC) condition, separated in 6 blocks. There was a maximum of 120 trials within each bloc. This gives the possibility to look at learning profiles (temporal evolution of judgement accuracy and/or task success

rate). Does inter-subject variability in learning profiles are related to specific strategies of delaying movement onsets/cancelling the ongoing movement, especially under tight temporal constraints? How early in the learning process individuals that have better success rates are able optimize their behavior?

- **RESPONSE:** Learning processes and inter-subject variability are important points in understanding our findings. We have added new results in Supplementary information (Supplementary Fig. 4) and address them in Discussion (p. 7, lines 210-214) as follow.

- *“Though each assessment variable was evaluated in every session (including the maximum number of 120 trials), we observed no clear temporal evolution within session for the task success rate, the decision accuracy and changes in movement strategic variables, such as delaying the movement onset or move-then-stop, except the goal rate (Supplementary Fig. 4). This indicates that learning within session occurred infrequently.”*

- Supplementary Fig. 4 shows the differences in each variable between the early and late trial bins in single sessions. We observed a partial learning effect within session. The results showed that there was a significant difference in the goal rate, i.e., the movement accuracy, at 0.4 s TTC in J task ($p = 0.031$), while there remained significant differences between 0.4 s TTC and others in the late trial bins. On the other hand, there was no learning effect on task success rate, judgment accuracy and changes in movement strategies, delaying the movement onset or move-then-stop. This suggests that participants cannot learn these aspects in such short periods.

We also examined the inter-subject variability as you questioned. The following figure (Fig. A) shows the relationship between the individual's goal rate, for which the learning process was observed, and the individual's movement strategies. There was no significant relation between the change in the movement onset and the change in the goal rate (Fig. Aa), but there was a significant negative correlation between the change in the move-then-stop rate and the change in the goal rate (Fig. Ab). However, the move-then-stop rate itself, on average, didn't show the learning effect, suggesting that the cancellation strategy less impacts the learning of the goal rate.

Figure A. Inter-subject variability in learning profiles. **a** Relation between the difference in the onset time between the early and late trial bins at 0.4s TTC and the goal rate. Each magenta circle shows each participant. The black dotted line shows the line where x-axis and y-axis are zero. r^2 represents the coefficient of determination. Pearson's correlation coefficient was -0.294 but there was no significant correlation ($p = 0.354$). **b** Relation between the difference in move-then-stop rate and that of the goal rate. Pearson's correlation coefficient was -0.622 and there was a significant negative correlation ($p = 0.0308$).

5. Major comment 4. Task-relevant resources. Judgement time is driven by the information that participants could process to initiate their behavior. In the present study, what information is considered to initiate these processes? It might be important to identify the task-relevant information and resources the agent has access to in order to solve the task.

- **RESPONSE:** We considered that the judgment and motor process would be comprehensively processed according to the time available because time proportion of movement onset time and judgment time to TTC was constant for TTC conditions (Supplementary Fig. 3c). We added this result in the Results (p.6, lines 168-169) and Discussion (p. 8, line 228) as follow.
 - *“We found that movement onset time, judgement time, and movement duration significantly decreased, in similar proportions, as TTC decreased (Supplementary Fig. 3a, b) (Mean \pm SD judgment time = 239 \pm 31.8, 305 \pm 40.5, 366 \pm 42.4 ms, onset time = 226 \pm 39.1, 283 \pm 57.8, 335 \pm 60.7 ms, movement duration time = 226 \pm 39.1, 283 \pm 57.8, 335 \pm 60.7 ms, for 0.4, 0.5 and 0.6 s TTC, respectively), indicating that the time ratio of decision-making*

to movement execution was constant regardless of the time constraint (Supplementary Fig. 3c).” in the Results (p.6, lines 168-169).

- *“Congruent with these studies, our results indicate that the time allocation between decision-making and motor execution is kept even for durations under one second (Supplementary Fig. 3c).”* in the Discussion (p.8, line 228).
- The decision as to whether target would pass through the strike area (Strike) or not (Ball) and the movement of where to reach could be made based on the task-relevant sensory information at given time (i.e., relative position of the target to the strike area) and accumulated evidence (i.e., final potential position predicted from target velocity and motion) in combination with other information such as strike area and hand cursor. We described these points in Discussion (p. 8, lines 235-243) as follows.
 - *“We also found that time proportion of movement onset and decision to TTC were constant among the TTC conditions examined (Supplementary Fig. 3c). A possible interpretation is that the movement onset as well as decision timing would be comprehensively processed according to the time constraint. These movement and decision could be made based on sensory information at a certain time (i.e., target position) or on urgency signal-related accumulated evidence (i.e., target velocity and motion) in combination with other information such as strike area and hand cursor. Participants are likely to utilize such task-relevant information to judge whether the target will pass through the strike area (Strike) or not (Ball) and to estimate the position to be reached.”*

6. Major comment 5. Embodied cognition/NoGo judgements under severe timing constraints. The results of No-Go judgement are very interesting and should be put forward for its full potential, i.e. for its implication on our understanding of the behavior. Tasks and skills can be trained to improve behavior. This can be underlined by mental representations, but there is a point when the mental representations are not as useful for skill development. In case of tight timing constraints, the skill becomes more instinctual and based on the experience of trial-to-trial feedback that update what information to rely on to efficiently link perception and action. With practice, participants are able to make judgements and decisions for the best possible outcome more quickly and efficiently based upon the affordances available but without having to take the time to consider all possible outcomes or consequences of each decision. As the system

evolves and grows, the outcome of these judgements and decisions add upon themselves and create a new set of initial conditions that lead to new judgement and decision-making behaviors. My question is linked to the previous comment (#3). What are the conditions in the present task that make the participant use the strategy go-stop? At which point along motor repetitions the consequences of one decision feedback onto itself to create an optimal set of judgements that lead to accurate decisions? Is this point occurring earlier or later during tight compared to larger timing constraints?

- **RESPONSE:** As you pointed out, when time constraints become severe, mental representation would be no longer useful, and indeed the responses can be explained by affordance alone. I have added a possible explanation as to how affordance could make the participant use the strategy go-stop and the mechanisms involved in go-stop in Discussion (p. 9, lines 275-296) as follows, and some references (52-63).

➤ *“Embodied decisions are made by competition between internal representations of potential action^{34,52,53}, which can be regarded as affordances defined as the perception of possibilities for, or restraints on, an action that the environment offers^{54,55}. These affordance competition hypotheses proposed that these affordances could be biased by the desirability of the expected outcome or the necessary effort^{46,56,57}. In light of the affordance competition hypothesis, our task could provide participants with an available affordance (a reachable strike area by moving the hand) and another affordance (a start position without moving the hand). The former could produce the expected affordances that those actions make available (i.e., hitting the target and returning it to the goal or resulting in FA), and the latter could produce the expected affordances (no hitting in CR). These expected affordances would impose a top-down bias in favor of hitting the target (a reachable strike area by moving the hand) when the time comes too short for accomplishing the task by initiating movement after the decision, which could cause change in movement strategy that the move rate increased as the time constraints became shorter. This is because task completion was to return the target to the goal area 20 times. We speculate that affordances could be biased with time constraints by urgency signals³⁶.*

Since our task involved motor inhibition, it triggered not only activation (Go) but also the inhibition (stop) strategy. The inhibitory process usually takes place about 100-200 ms after the onset of the stop stimulus⁵⁸⁻⁶⁰.

Current models of inhibitory control propose that the processes of inducing ("Go") and withdrawing ("Stop") the behavior are nearly independent, ultimately interacting to inhibit potential or actual behavioral responses when a stop signal is externally presented⁶¹⁻⁶³."

- On the other hand, as shown by Fig. 3, we found a correlation between the judgment time and the movement onset time, which also correlates with judgment accuracy. However, as mentioned above, Supplementary Fig. 4d shows no time evolution in movement onset time within the session. Thus, we speculate that an optimal set of judgments that lead to accurate decisions would be created by each individual from the beginning and probably not acquired during short-term repetitions.

7. Minor comment 1. Introduction: lines 62-63, however is repeated twice. Double opposition makes the reasoning difficult to follow.

- **RESPONSE:** I have removed an unnecessary sentence and left only the first sentence (p. 3, lines 73-75) as follows. We hope that the edited part clarifies that the movement pattern itself could be changed.

➤ *"However, when the time constraint was less than 0.5 s they applied the strategy of changing movement duration rather than movement onset²⁴."*

8. Minor comment 2. Population: Since expertise plays a role in decision making regarding rapid movements, how expert were the participants in regards to baseball (occasional players or naïve versus regular players)? Were the participants paid for their participation?

- **RESPONSE:** Two participants had baseball experience, but the rest were novices. We didn't focus on sport expertise in this study. Participants were not paid because we had employees volunteer to participate.

9. Minor comment 3. Results. The variables that were submitted to the statistical analyses were not all introduced. For instance, results of judgement time are presented in the second section of the Results. While the reader can guess what movement time is, judgement time deserves a quick explanation without having to dig into the supplemental material, for a matter of fluidity in the reading process. Maybe a figure of hand movement profile would help to dissociate or make the link between the different variables, including the judgement time.

- **RESPONSE:** I apologize for the insufficient introduction of the statistical analyses. As you suggested, we have moved the figure showing the hand movement profile from the supplement to the main figure (Fig. 1c) (p.21, lines 666-670), and explained how we calculated the judgment time and movement onset time and those values at the third paragraph in the Results (p.5, lines 160-163, p.6, lines 165-168) as follows.

- *“**c** Example of hand acceleration profiles in the forward (y) direction at 0.5 s TTC for a representative participant. **c** Judgment times (blue circles) were defined as the time when each Hit trial acceleration (gray line) diverged from the average acceleration for all No-go (CR and Miss) trials (red line) in J task. Each CR or Miss trial is indicated by a dashed red line. Movement onset times are shown by cyan circles.”* in the caption of the main figure (Fig. 1c) (p.21, lines 666-670).
- *“Thus, we evaluated some temporal features, judgment time, movement onset time and movement duration, based on the hand acceleration profiles in the forward (y) direction in Hit trials (see Methods and Fig. 1c and Supplementary Fig. 1d). The judgment time was defined as the time at which the Hit trial diverged from the average No-go trial. The movement onset time was determined as the time when the Y direction acceleration exceeded 10% of its maximum value. The movement duration was defined as the time from onset time to target contact time.”* in the Results (p.5, lines 160-163, p.6, lines 165-168).

10. Minor comment 4. Discussion lines-199-200. “Our result suggests that movement onset might force decision completion.” This statement is difficult to understand when we consider that for the 0.4 s condition, movements were initiated and stopped, which was correlated with judgement success. This rather suggests that the relationship between movement onset and the decision completion is more complex and depends on the timing constraints: Movement onset can also occur independently from the final decision to move or not to move in situation when timing constraints are severe.

- **RESPONSE:** As you pointed out, movement onset can occur independently from the final decision; the decision is frequently delayed when timing constraints are severe. Thus, the description that movement onset might force decision completion is inappropriate. We have revised it to “Our result suggests that movement and decision processes occurred concurrently” (p.8, lines 251-252).

In addition to the pointers, we have corrected the wording to be more appropriate and fixed wrong words and variable values in Supplementary Information.

Responses to Reviewer #2

Thank you for the time and effort expended in generating the thoughtful and constructive feedback on our manuscript. Our responses to your comments are as follows. Corrected and added parts are noted in red in the manuscript and the Supplementary information.

Sincerely yours,

Akemi Kobayashi

1. **Main Comment:** My impressions is that this paper deals with a very interesting topic, which is the relationship between decision-making between actions and motor control. Furthermore, under the interesting constraint of severe time pressure. I also believe the paper provides interesting results, in so far 0.5s seems to be the boundary to trade-off between decision-making and motor control in their current framework.

However, the general impression is that the paper requires some additional general work. It lacks structure, the proper introduction of hypotheses, the experiments are not properly introduced. It is in general difficult to read and therefore hard to understand. I cite a few of the problems next:

- **RESPONSE:** We have revised to properly and intelligibly explain the points that you pointed out as follows.
2. **Problem 1:** The notion of TTC is crucial in the experiment, and is not properly described until very late in the paper.
 - **RESPONSE:** We have added the description of TTC in the Introduction (p. 3, line 66) for early understanding as follows.
 - *“In this task, the time from target start to reach contact (Time-to-contact; TTC) was restricted to 0.4s, 0.5s, or 0.6s.”*
 3. **Problem 2:** Line 87. What is start area? It is nowhere to be seen in FIG 1.
 - **RESPONSE:** I apologize for the difficulty in seeing the figure. The red circle at the initial position of the hand in Fig. 1 indicates the start area. We have changed the position of the handle to make it easier to see.
 4. **Problem 3:** Lines 89-90. The description of the Strike vs Ball targets is first demands a little more introduction. Presentation of both tasks is difficult to understand because it is

not properly introduced. Why the two tasks, and why in this specific fashion? In fact, I miss the clear formulation of a hypothesis that justifies and guides this particular experimental arrangement.

- **RESPONSE:** We agree with you. The explanation of the task and its characteristics was inadequate and we have rewritten it in the Introduction (p. 3, lines 82, 85-91) as follows.
 - *“The present study examines how decision-making and motor execution/regulation are achieved under severe time constraints using a baseball-like fast-hitting paradigm with Go (Strike) or No-go (Ball) decision-making. Participants are required to hit a moving target in the strike area with a hand cursor and return the target toward a frontal goal area, but to refrain from hitting the Ball targets (Fig. 1a). The Strike targets are more centered and consequently expected to be easier to hit than the Ball targets, which provides an advantage in making Go/No-go judgments. However, this task had a trade-off between judgment and movement accuracy. The participants must balance this trade-off in a limited time so as to enhance task performance. To assess judgment efficacy, participants also perform a hitting-only task without Go/No-go decision-making (NJ task) to compare to the hitting task with Go/No-go decision-making (J task).”*
- We designed this task because our interest is in rapid sensorimotor decision when the motor demands are high, such as in sports, and in hitting a target; the decision to hit or not to hit is important. We also elaborated our hypothesis in the Introduction (p.4, lines 95-98) as follows.
 - *“We hypothesize that the Go/No-go judgment is effective on task performance, but the efficacy is reduced in short time constraints because of a trade-off between judgment and movement accuracies. However, movement strategy is changed so as to compensate for the reduction in judgment efficacy.”*

5. **Problem 4:** Line 96. What is d' ? as a judgement accuracy?

- **RESPONSE:** I apologize for the insufficient explanation. We have replaced " d " with "*sensitivity d' , a statistical measure to quantify how a system distinguishes signal distribution from noise distribution*" (p. 4, lines 116, 119-120) as follows. It indicates how well participants separate the correct signal distribution from noise distribution (in our task, Strike or Ball).

- “Using hand movement profiles, we classified the observed movements into four possible *judgment* responses based on signal detection theory²⁹, i.e., Hit (Go in Strike), Miss (No-go in Strike), False Alarm (FA; Go in Ball), and Correct Rejection (CR; No-go in Ball) (see Supplementary Fig. 1a, b). As the judgment accuracy, we assessed *sensitivity d'*, a statistical measure to quantify how a system distinguishes signal distribution from noise distribution, for each TTC by calculating the ratio of four responses, Hit, Miss, FA, and CR.”
6. **Problem 5:** From Suppl FIG 1B it looks as if all No-Go targets are in the plus-minus 15 degrees orientation. That would make the predictable. Does this carry any implication for their conclusions?
- **RESPONSE:** The targets were selected pseudo-randomly and the participants could not predict them. In our task, the Ball target was only ± 15 degrees, but in a real batting scenario, the variability of possible locations is large and some courses are more difficult than ours, so the decision-making process can be expected to be more difficult and severe.
7. **Problem 6:** Supl. FIG 1C-D, align these curves on movement onset, and make that the origin of time. They're hard to read just like that.
- **RESPONSE:** We apologize for the difficulty. We have added a new Supplementary Fig. 1c, e that aligns the data from Supplementary Fig. 1c-d (p.14, line 443) on movement onset, and made that the origin of time.
 - “Using the Y direction acceleration data in J task, the judgment time for Hit responses was determined as the time when the value obtained by subtracting the average data of all No-go (CR and Miss) trials from each Hit response data exceeded the last time the threshold value was exceeded (Fig. 1c and Supplementary Fig. 1c, d, e).” (p.14, line 443).
8. **Problem 7:** The statistics must provide accurate p-values. Providing boundaries ($p < 0.01$ or $p < 0.05$) is not good enough.
- **RESPONSE:** We have replaced boundaries with accurate p-values except description as $p < 0.0001$ when the p-value is less than 0.0001 (lines 130, 131, 133, 141, 143, 150, 171, 173, 174, 198) as follows.
 - “A two-way analysis of variance (2-way ANOVA) showed significant main effects of TTC ($F(2, 66) = 35.22, p < 0.0001$) and task ($F(1, 66) = 29.90, p <$

0.0001); there was no significant interaction ($F(2, 66) = 3.58, p = 0.033$). The Post-hoc Bonferroni-corrected t -tests showed significant differences between tasks with 0.5 s ($p = 0.0078$) and 0.6 s TTC ($p = 0.00019$), but none at 0.4 s TTC ($p = 0.97$).” (lines 129-133).

- “We found a significant reduction in *the goal rate* (correct hitting rate to Strike trials) for J task at 0.4 s TTC compared with *the goal rate for J task* ($p = 0.010$, Fig. 2b), although there was no significant main effect of task ($F(1, 66) = 2.06, p = 0.156$) and significant interaction ($F(2, 66) = 5.49, p = 0.0063$),” (lines 140-143).
- “We found that judgment accuracy decreased significantly as TTC fell ($F(2, 33) = 39.77, p < 0.0001$), with a pronounced reduction from 0.5 s to 0.4 s TTC ($p < 0.0001$).” (lines 149-150).
- “although there was a significant difference ($F(1, 66) = 4.05, p = 0.048$) (Supplementary Fig. 3a). Indeed, we found a strong association between movement onset time and judgment time ($r^2=0.863, p < 0.0001$, Fig. 3a). In addition, movement onset time was significantly correlated with judgement accuracy ($r^2=0.372, p < 0.0001$, Fig. 3b).” (lines 170-174).
- “We found a significant correlation between the success rate of move-then-stop movements and judgement accuracy (d') at 0.4 TTC ($r^2=0.69, p = 0.0083$, Fig. 4c),” (lines 197-198).
- In addition, the exact description of the p -value in the figure has been added (p.5, lines 133-136, 144-148, 185) as follows.
 - “With regard to the task J success rate, there were significant differences between 0.4 s and 0.6 s TTC ($p < 0.0001$), and between 0.4 s and 0.5 s TTC ($p < 0.0001$). As for the NJ task success rate, there were significant differences between 0.4 s and 0.6 s TTC ($p = 0.0038$).” (lines 133-136).
 - “With regard to the goal rate for J task, the Post-hoc Bonferroni-corrected t -tests showed significant differences between 0.4 s and 0.6 s TTC ($p < 0.0001$), between 0.4 s and 0.5 s TTC ($p < 0.0001$), and between 0.5 s and 0.6 s TTC ($p = 0.038$). As for the goal rate for J task, there were significant differences between 0.4 s and 0.6 s TTC ($p = 0.023$).” (lines 144-148).

- “we found that participants were apt to move their hand even in No-go trials as TTC fell ($F(2, 33) = 3.35, p = 0.047$, Fig. 4a, b).” (lines 184-185).
- Some of the F-values were wrong, so I have corrected them as follow (p.5, lines 130, 131). The suggestions from the results are not affected by this.
- “A two-way analysis of variance (2-way ANOVA) showed significant main effects of TTC ($F(2, 66) = 35.22, p < 0.0001$) and task ($F(1, 66) = 29.90, p < 0.0001$); there was no significant interaction ($F(2, 66) = 3.58, p = 0.033$).”

9. **Problem 8:** There is a number of papers providing some interesting insight onto the SAT they mention. The authors should consider basing their argument on this previous work. From the top of my head Reynaud et al. 2020. They already start dealing with time constraints, as does as well the Thura et al. 2014 you cite.

- **RESPONSE:** Thank you for providing these important insights. I apologize for my lack of survey coverage. As you suggested, we have added this work, Reynaud et al. 2020 et al. in the Introduction (p.3, lines 55-59) and in the Discussion (p.7, lines 221-226) as follows.
- “Reynaud et al.¹⁹ have revealed the trade-off between decision making and motor process according to time constraints of up to a few seconds by manipulating motor cost for a responding target in a token task. Their results indicate that fast and inaccurate decisions were often made before more demanding movements so that participants sacrificed decision making for action execution.” in the Introduction (p.3, lines 55-59).
- “Other studies have argued that urgent signals, assumed to be temporally strengthening signals related to the urgency in making a decision³⁶, increased the speed of triggering reaching movement and the vigor of saccades^{17,37}, suggesting a *shared* mechanism between decision-making and motor processes that involve the basal ganglia^{17,37,38}. A recent study on manipulating motor contexts showed that slow movements executed for demanding targets were often preceded by faster and less accurate decisions, indicating that participants sacrificed decision-making for action execution¹⁹. This suggests that decision urgency and movement vigor signals are likely to interact.” in the Discussion (p.7, lines 221-226).
- We hope these revisions provide a more thorough discussion.

10. **Problem 9:** In the discussion, the authors mention that movement onset might force decision completion. The view that a decision can be finished before the movement has been completed cannot be qualified as embodied.
- **RESPONSE:** As you pointed out, the description that movement onset might force decision completion is inappropriate as embodied. We have revised to “*Our result suggests that movement and decision processes occurred concurrently*” (p. 8, lines 251-252).
11. **Problem 10:** The definition of success rate across the three task is not homogeneous, as you are quantifying different metrics across different experiments. I would be cautious to call all of this success rate on equal terms.
- **RESPONSE:** We agree that it was confusing to use the same word “*success rate*” for different metrics and have replaced the term “*success rate*” throughout the paper with “*NJ task success rate*”, “*J task success rate*”, and “*Goal rate for Strike targets*” to use more precise terms. To reflect this, we have revised the main figure (Fig. 1a, b) and Supplementary Fig. 2 and description in the Results (lines 121, 122-124, 126-128, 133-136, 140, 141, 144-147, 151, 152) and in the Data analysis (lines 429, 430, 432-436, 438,439) and in the caption of Fig. 2 (p.21, lines 674-676, 678, 680) as follows.
 - “*The Hit responses were further divided into two responses according to goal success, e.g., Hit-Goal and Hit-No Goal. We assessed goal rates (correct hitting; Hit-Goal rate to all Strike trials) and overall task success rate for J task, the J task success rate, which was defined as the ratio of the sum of the correct hitting (Hit-Goal) and the correct judgment (CR) responses in all trials (see Methods). Corresponding task success rates for NJ task, NJ task success rate, were also evaluated as the ratio of the correct hitting.*

Both the J task success rate and the NJ task success rate decreased with shorter TTCs, but the trends differed between tasks (Fig. 2a).” (lines 121-129).

 - “*With regard to the task J success rate, there were significant differences between 0.4 s and 0.6 s TTC ($p < 0.0001$), and between 0.4 s and 0.5 s TTC ($p < 0.0001$). As for the NJ task success rate, there were significant differences between 0.4 s and 0.6 s TTC ($p = 0.0038$).*” (lines 133-136).

- “We found a significant reduction in *the goal rate* (correct hitting rate to Strike trials) for J task at 0.4 s TTC compared with *the goal rate for J task* ($p = 0.010$, Fig. 2b),” (lines 140-142).
- “With regard to the goal rate for J task, the Post-hoc Bonferroni-corrected t-tests showed significant differences between 0.4 s and 0.6 s TTC ($p < 0.0001$), between 0.4 s and 0.5 s TTC ($p < 0.0001$), and between 0.5 s and 0.6 s TTC ($p = 0.038$). As for the goal rate for J task, there were significant differences between 0.4 s and 0.6 s TTC ($p = 0.023$).” (lines 144-148).
- “The reduction in *the J task success rate* was mainly caused by lower judgement accuracy, since *the NJ task success* barely changed from 0.5 to 0.4 TTC (Fig. 2a).” (lines 151-152).
- “We defined *the J task success rate and the NJ task success rate, which regarded as total task performance, and the goal rate for Strike targets*, as follows (see also Supplementary Fig. 1b).

$$J \text{ task success rate (Strike + Ball)} = \frac{\text{Hit-Goal} + \text{CR}}{\text{Total}}$$

$$NJ \text{ task success rate (Strike + Ball)} = \frac{\text{Goal}}{\text{Total}}$$

$$goal \text{ rate (Strike only)} = \frac{\text{Strike-Goal}}{\text{Strike}}$$

The *J task success rate* was defined as the number of Hit-Goal and CR responses divided by the total number of trials, and *the NJ task success rate* was defined as the number of goal responses divided by the total number of trials. In order to examine the impact of judgment on hitting accuracy, we determined the *goal rate* (Strike). It was defined as the number of Strike-Goal responses divided by the number of Strike trials for J task and NJ task.” in the Data analysis (lines 429-439).

- “**a, b** The mean *task success rate* for Strike and Ball targets (**a**) and *the mean goal rate* for only Strike targets (**b**) for each TTC are plotted as orange lines (*the J task success rate*) and blue lines (*the NJ task success rate*) (Mean \pm SEM, * $p < 0.05$, ** $p < 0.01$). **c** The mean judgement accuracy (d') for each TTC is shown by orange lines (Mean \pm SEM, * $p < 0.05$, ** $p < 0.01$) with individual data points overlaid. Please note that *the J task success rate* and

judgment accuracy decreased as TTCs were shortened, and judgment was ineffective in terms of J task success rate if TTC was less than 0.5 seconds.”
in the caption of Fig. 2 (p.21, lines 673-680).

In addition to the pointers, we have corrected the wording to be more appropriate and fixed wrong words and variable values in Supplementary Information.

REVIEWERS' COMMENTS:

Reviewer #1 (Remarks to the Author):

The authors provided clear and convincing arguments to all of the major and minor points. I thank the authors for their open-mindedness and constructive reception of the suggestions. I have no further question.

Reviewer #2 (Remarks to the Author):

I just finished reading the authors's new version of the article, their rebuttal and the list of changes, and I am pleased to say that the authors have taken into consideration every single aspect raised during the first review round and modified the manuscript accordingly. I have no further reservations as to its suitability for publication.

Rebuttal Letter

Nov 22th, 2021

Responses to Reviewer #1

Thank you for the time and effort expended in generating the thoughtful and constructive feedback on our manuscript. Our responses to your comments are as follows. Corrected and added parts are noted in red in the manuscript and the Supplementary information.

Sincerely yours,

Akemi Kobayashi

2. Major comment 1. A priori hypothesis. There is no clear definition of the tested hypothesis in the introduction. If the study was exploratory, the authors should make a clear mention of that fact, or introduce earlier studies. For instance, it seems that the concept of visuomotor delay could play a role in the present study. Visuomotor delay is The duration of visuomotor delays appears to depend on the subsequent manner in which the information is used. It can be as short as 100 ms when information is used during the on-line regulation of the movement (Bootsma and van Wieringen, 1990; Savelsbergh et al., 1991), but can reach values near 200 ms when information is used to produce movement relying on decision processes or more discrete movements, such as the beginning of the movement or some important correction of that movement (Lee et al., 1983 ; Michaels et al., 2001 ; Benguigui et al., 2003). A study evaluated the effect of shorter latencies of time-to-contact on hand movement profiles when hitting balls in a group of non-expert as well as expert tennis players (Le Runigo, Bardy and Benguigui, 2005 Human Movement Science). Participants had to adapt their movements by making decisions (increase or decrease hand movement velocities) at 0.4 s and 0.26 s before time to contact, in delays compatible with the present study. This could represent a rationale to build hypotheses about embodied decision and movement kinematics under tight timing constraints.

If needed, these references are in:

- **RESPONSE:** As the reviewer pointed points out, visuomotor delay appears to be a reasonable basis on which to build our hypotheses. The Introduction (p.3, lines 62-69, 75-81) and Discussion (p.8, lines 266-267) now reference previous studies

on decision-making and visuomotor delay to explain the relationship between decision-making and motor process in terms of time constraints as follows.

- *“Severe time constraints are expected to impact both decision-making and motor execution. Regarding decision-making, Owens et al.²² assessed a baseball batting-specific Go/No-go decision responses, in which participants were required to reach the screen if the moving target passed within a certain area and not to reach if it passed outside the area. In this task, the time from target start to reach contact (Time-to-contact; TTC) was restricted to 0.4s, 0.5s, or 0.6s. Result showed that the decision accuracy decreased significantly when TTC reached 0.4s, but made no mention of the motor execution aspects such as movement duration and trajectory. Movement strategy can be also changed with time constraints.”* in the Introduction (p.3, lines 62-69).
- *“Visuomotor delay, which is defined as the time between the emergence of visually detectable information and the initiation of resulting motor adjustment²⁵, is a possible determinant of sensorimotor decision behavior. The extent of this delay has been shown to vary between 100ms and 300ms depending on the ongoing task, such as the online regulation of the movement²⁶ and the initiation of movement²⁷, and motor expertise²⁸. While these studies indicate that short time constraints affect decision-making and motor execution, it is unclear how they are balanced under severe time constraints.”* in the Introduction (p.3, lines 75-81).
- *“Shorter visuomotor delay in continuous movement compared to discrete movement⁴⁹ can be also related with change to ongoing go-stop strategy.”* in the Discussion (p.8, lines 266-267).
- We now also describe our hypothesis briefly in the last paragraph of the Introduction (p.4, lines 95-98) as follows.
 - *“We hypothesize that the Go/No-go judgment is effective on task performance, but the efficacy is reduced in short time constraints because of a trade-off between judgment and movement accuracies. However, movement strategy is changed so as to compensate for the reduction in judgment efficacy.”*

3. Major comment 2. Eye-movement kinematics. Many decisions such as to swing at a baseball pitch are accompanied by characteristic eye-movement behavior, preceding the onset of hand movements. Using a Go/NoGo paradigm, Go/NoGo decisions were

reflected in distinct eye-movement responses (Fooker and Spering, 2019; Journal of Vision). Under tight time constraints between the moment when the ball appears in the visual field of view and the moment when one has to initiate a hand movement or not, decision making relies on the speed with which visual information can be detected, and might depend on eye-hand coordination. Did the authors measure eye-movement kinematics? Could eye-movement kinematics help to refine the link between decision signals and hand motor actions?

- **RESPONSE:** We didn't measure eye-movement kinematics because of limitations in the experimental design and measurements. However, as you mentioned, eye-movement kinematics and eye-hand coordination could refine the link between decision signals and hand motor actions. We discussed the role of eye-movement behavior in interceptive tasks in the Discussion (p.10, lines 303-315) as follows.
 - *“Our research has some limitations that should be addressed. Many studies have shown common or coordinated control of eye and hand movements in interceptive tasks, but we did not measure eye movements in this study. Fooker and Spering⁶⁹ showed that smooth pursuit and saccadic eye movement parameters provide reliable estimates of Go/No-go manual intercept. This indicates that eye movement modulations reflect an early readout of decision formation itself. Owens et al²² also designed a baseball-specific Go/No-go task with screen touch under a time constraint of 0.4s to 0.6s comparable to our study, and showed that baseball experts showed better gaze control than novices at the short preparation and fast stimulus speed trials (0.4s TTC) including significant differences in stimulus tracking, gaze to stimulus distance, and peak pupil dilation. Whereas saccade onset, hand movements, and screen touch reaction times did not differ between groups. Additional research is needed to clarify how eye-movement kinematics relate to decision making and motor execution under the time constraint.”*

4. Major comment 3. Learning processes and inter-subject variability. Participants performed consecutive trials for each Time To Contact (TTC) condition, separated in 6 blocks. There was a maximum of 120 trials within each bloc. This gives the possibility to look at learning profiles (temporal evolution of judgement accuracy and/or task success rate). Does inter-subject variability in learning profiles are related to specific strategies of delaying movement onsets/cancelling the ongoing movement, especially under tight

temporal constraints? How early in the learning process individuals that have better success rates are able to optimize their behavior?

- **RESPONSE:** Learning processes and inter-subject variability are important points in understanding our findings. We have added new results in Supplementary information (Supplementary Fig. 4) and address them in Discussion (p. 7, lines 210-214) as follows.
 - *“Though each assessment variable was evaluated in every session (including the maximum number of 120 trials), we observed no clear temporal evolution within session for the task success rate, the decision accuracy and changes in movement strategic variables, such as delaying the movement onset or move-then-stop, except the goal rate (Supplementary Fig. 4). This indicates that learning within session occurred infrequently.”*
- Supplementary Fig. 4 shows the differences in each variable between the early and late trial bins in single sessions. We observed a partial learning effect within session. The results showed that there was a significant difference in the goal rate, i.e., the movement accuracy, at 0.4 s TTC in J task ($p = 0.031$), while there remained significant differences between 0.4 s TTC and others in the late trial bins. On the other hand, there was no learning effect on task success rate, judgment accuracy and changes in movement strategies, delaying the movement onset or move-then-stop. This suggests that participants cannot learn these aspects in such short periods.

We also examined the inter-subject variability as you questioned. The following figure (Fig. A) shows the relationship between the individual's goal rate, for which the learning process was observed, and the individual's movement strategies. There was no significant relation between the change in the movement onset and the change in the goal rate (Fig. Aa), but there was a significant negative correlation between the change in the move-then-stop rate and the change in the goal rate (Fig. Ab). However, the move-then-stop rate itself, on average, didn't show the learning effect, suggesting that the cancellation strategy less impacts the learning of the goal rate.

Figure A. Inter-subject variability in learning profiles. **a** Relation between the difference in the onset time between the early and late trial bins at 0.4s TTC and the goal rate. Each magenta circle shows each participant. The black dotted line shows the line where x-axis and y-axis are zero. r^2 represents the coefficient of determination. Pearson's correlation coefficient was -0.294 but there was no significant correlation ($p = 0.354$). **b** Relation between the difference in move-then-stop rate and that of the goal rate. Pearson's correlation coefficient was -0.622 and there was a significant negative correlation ($p = 0.0308$).

5. Major comment 4. Task-relevant resources. Judgement time is driven by the information that participants could process to initiate their behavior. In the present study, what information is considered to initiate these processes? It might be important to identify the task-relevant information and resources the agent has access to in order to solve the task.

- **RESPONSE:** We considered that the judgment and motor process would be comprehensively processed according to the time available because time proportion of movement onset time and judgment time to TTC was constant for TTC conditions (Supplementary Fig. 3c). We added this result in the Results (p.6, lines 168-169) and Discussion (p. 8, line 228) as follow.
 - *“We found that movement onset time, judgement time, and movement duration significantly decreased, in similar proportions, as TTC decreased (Supplementary Fig. 3a, b) (Mean \pm SD judgment time = 239 ± 31.8 , 305 ± 40.5 , 366 ± 42.4 ms, onset time = 226 ± 39.1 , 283 ± 57.8 , 335 ± 60.7 ms, movement duration time = 226 ± 39.1 , 283 ± 57.8 , 335 ± 60.7 ms, for 0.4, 0.5 and 0.6 s TTC, respectively), indicating that the time ratio of decision-making*

to movement execution was constant regardless of the time constraint (Supplementary Fig. 3c).” in the Results (p.6, lines 168-169).

- *“Congruent with these studies, our results indicate that the time allocation between decision-making and motor execution is kept even for durations under one second (Supplementary Fig. 3c).”* in the Discussion (p.8, line 228).
- The decision as to whether target would pass through the strike area (Strike) or not (Ball) and the movement of where to reach could be made based on the task-relevant sensory information at given time (i.e., relative position of the target to the strike area) and accumulated evidence (i.e., final potential position predicted from target velocity and motion) in combination with other information such as strike area and hand cursor. We described these points in Discussion (p. 8, lines 235-243) as follows.
 - *“We also found that time proportion of movement onset and decision to TTC were constant among the TTC conditions examined (Supplementary Fig. 3c). A possible interpretation is that the movement onset as well as decision timing would be comprehensively processed according to the time constraint. These movement and decision could be made based on sensory information at a certain time (i.e., target position) or on urgency signal-related accumulated evidence (i.e., target velocity and motion) in combination with other information such as strike area and hand cursor. Participants are likely to utilize such task-relevant information to judge whether the target will pass through the strike area (Strike) or not (Ball) and to estimate the position to be reached.”*

6. Major comment 5. Embodied cognition/NoGo judgements under severe timing constraints. The results of No-Go judgement are very interesting and should be put forward for its full potential, i.e. for its implication on our understanding of the behavior. Tasks and skills can be trained to improve behavior. This can be underlined by mental representations, but there is a point when the mental representations are not as useful for skill development. In case of tight timing constraints, the skill becomes more instinctual and based on the experience of trial-to-trial feedback that update what information to rely on to efficiently link perception and action. With practice, participants are able to make judgements and decisions for the best possible outcome more quickly and efficiently based upon the affordances available but without having to take the time to consider all possible outcomes or consequences of each decision. As the system

evolves and grows, the outcome of these judgements and decisions add upon themselves and create a new set of initial conditions that lead to new judgement and decision-making behaviors. My question is linked to the previous comment (#3). What are the conditions in the present task that make the participant use the strategy go-stop? At which point along motor repetitions the consequences of one decision feedback onto itself to create an optimal set of judgements that lead to accurate decisions? Is this point occurring earlier or later during tight compared to larger timing constraints?

- **RESPONSE:** As you pointed out, when time constraints become severe, mental representation would be no longer useful, and indeed the responses can be explained by affordance alone. I have added a possible explanation as to how affordance could make the participant use the strategy go-stop and the mechanisms involved in go-stop in Discussion (p. 9, lines 275-296) as follows, and some references (52-63).

➤ *“Embodied decisions are made by competition between internal representations of potential action^{34,52,53}, which can be regarded as affordances defined as the perception of possibilities for, or restraints on, an action that the environment offers^{54,55}. These affordance competition hypotheses proposed that these affordances could be biased by the desirability of the expected outcome or the necessary effort^{46,56,57}. In light of the affordance competition hypothesis, our task could provide participants with an available affordance (a reachable strike area by moving the hand) and another affordance (a start position without moving the hand). The former could produce the expected affordances that those actions make available (i.e., hitting the target and returning it to the goal or resulting in FA), and the latter could produce the expected affordances (no hitting in CR). These expected affordances would impose a top-down bias in favor of hitting the target (a reachable strike area by moving the hand) when the time comes too short for accomplishing the task by initiating movement after the decision, which could cause change in movement strategy that the move rate increased as the time constraints became shorter. This is because task completion was to return the target to the goal area 20 times. We speculate that affordances could be biased with time constraints by urgency signals³⁶.*

Since our task involved motor inhibition, it triggered not only activation (Go) but also the inhibition (stop) strategy. The inhibitory process usually takes

place about 100-200 ms after the onset of the stop stimulus⁵⁸⁻⁶⁰. Current models of inhibitory control propose that the processes of inducing ("Go") and withdrawing ("Stop") the behavior are nearly independent, ultimately interacting to inhibit potential or actual behavioral responses when a stop signal is externally presented⁶¹⁻⁶³."

- On the other hand, as shown by Fig. 3, we found a correlation between the judgment time and the movement onset time, which also correlates with judgment accuracy. However, as mentioned above, Supplementary Fig. 4d shows no time evolution in movement onset time within the session. Thus, we speculate that an optimal set of judgments that lead to accurate decisions would be created by each individual from the beginning and probably not acquired during short-term repetitions.

7. Minor comment 1. Introduction: lines 62-63, however is repeated twice. Double opposition makes the reasoning difficult to follow.

- **RESPONSE:** I have removed an unnecessary sentence and left only the first sentence (p. 3, lines 73-75) as follows. We hope that the edited part clarifies that the movement pattern itself could be changed.

➤ *"However, when the time constraint was less than 0.5 s they applied the strategy of changing movement duration rather than movement onset²⁴."*

8. Minor comment 2. Population: Since expertise plays a role in decision making regarding rapid movements, how expert were the participants in regards to baseball (occasional players or naïve versus regular players)? Were the participants paid for their participation?

- **RESPONSE:** Two participants had baseball experience, but the rest were novices. We didn't focus on sport expertise in this study. Participants were not paid because we had employees volunteer to participate.

9. Minor comment 3. Results. The variables that were submitted to the statistical analyses were not all introduced. For instance, results of judgement time are presented in the second section of the Results. While the reader can guess what movement time is, judgement time deserves a quick explanation without having to dig into the supplemental material, for a matter of fluidity in the reading process. Maybe a figure of hand movement profile would help to dissociate or make the link between the different variables, including the judgement time.

- **RESPONSE:** I apologize for the insufficient introduction of the statistical analyses. As you suggested, we have moved the figure showing the hand movement profile from the supplement to the main figure (Fig. 1c) (p.21, lines 666-670), and explained how we calculated the judgment time and movement onset time and those values at the third paragraph in the Results (p.5, lines 160-163, p.6, lines 165-168) as follows.

➤ *“c Example of hand acceleration profiles in the forward (y) direction at 0.5 s TTC for a representative participant. c Judgment times (blue circles) were defined as the time when each Hit trial acceleration (gray line) diverged from the average acceleration for all No-go (CR and Miss) trials (red line) in J task. Each CR or Miss trial is indicated by a dashed red line. Movement onset times are shown by cyan circles.”* in the caption of the main figure (Fig. 1c) (p.21, lines 666-670).

➤ *“Thus, we evaluated some temporal features, judgment time, movement onset time and movement duration, based on the hand acceleration profiles in the forward (y) direction in Hit trials (see Methods and Fig. 1c and Supplementary Fig. 1d). The judgment time was defined as the time at which the Hit trial diverged from the average No-go trial. The movement onset time was determined as the time when the Y direction acceleration exceeded 10% of its maximum value. The movement duration was defined as the time from onset time to target contact time.”* in the Results (p.5, lines 160-163, p.6, lines 165-168).

10. Minor comment 4. Discussion lines-199-200. “Our result suggests that movement onset might force decision completion.” This statement is difficult to understand when we consider that for the 0.4 s condition, movements were initiated and stopped, which was correlated with judgement success. This rather suggests that the relationship between movement onset and the decision completion is more complex and depends on the timing constraints: Movement onset can also occur independently from the final decision to move or not to move in situation when timing constraints are severe.

- **RESPONSE:** As you pointed out, movement onset can occur independently from the final decision; the decision is frequently delayed when timing constraints are severe. Thus, the description that movement onset might force decision completion is inappropriate. We have revised it to *“Our result suggests that movement and decision processes occurred concurrently”* (p.8, lines 251-252).

In addition to the pointers, we have corrected the wording to be more appropriate and fixed wrong words and variable values in Supplementary Information.

Responses to Reviewer #2

Thank you for the time and effort expended in generating the thoughtful and constructive feedback on our manuscript. Our responses to your comments are as follows. Corrected and added parts are noted in red in the manuscript and the Supplementary information.

Sincerely yours,

Akemi Kobayashi

1. **Main Comment:** My impressions is that this paper deals with a very interesting topic, which is the relationship between decision-making between actions and motor control. Furthermore, under the interesting constraint of severe time pressure. I also believe the paper provides interesting results, in so far 0.5s seems to be the boundary to trade-off between decision-making and motor control in their current framework.

However, the general impression is that the paper requires some additional general work. It lacks structure, the proper introduction of hypotheses, the experiments are not properly introduced. It is in general difficult to read and therefore hard to understand. I cite a few of the problems next:

- **RESPONSE:** We have revised to properly and intelligibly explain the points that you pointed out as follows.
2. **Problem 1:** The notion of TTC is crucial in the experiment, and is not properly described until very late in the paper.
 - **RESPONSE:** We have added the description of TTC in the Introduction (p. 3, line 66) for early understanding as follows.
 - *“In this task, the time from target start to reach contact (Time-to-contact; TTC) was restricted to 0.4s, 0.5s, or 0.6s.”*
 3. **Problem 2:** Line 87. What is start area? It is nowhere to be seen in FIG 1.
 - **RESPONSE:** I apologize for the difficulty in seeing the figure. The red circle at the initial position of the hand in Fig. 1 indicates the start area. We have changed the position of the handle to make it easier to see.
 4. **Problem 3:** Lines 89-90. The description of the Strike vs Ball targets is first demands a little more introduction. Presentation of both tasks is difficult to understand because it is

not properly introduced. Why the two tasks, and why in this specific fashion? In fact, I miss the clear formulation of a hypothesis that justifies and guides this particular experimental arrangement.

- **RESPONSE:** We agree with you. The explanation of the task and its characteristics was inadequate and we have rewritten it in the Introduction (p. 3, lines 82, 85-91) as follows.
 - *"The present study examines how decision-making and motor execution/regulation are achieved under severe time constraints using a baseball-like fast-hitting paradigm with Go (Strike) or No-go (Ball) decision-making. Participants are required to hit a moving target in the strike area with a hand cursor and return the target toward a frontal goal area, but to refrain from hitting the Ball targets (Fig. 1a). The Strike targets are more centered and consequently expected to be easier to hit than the Ball targets, which provides an advantage in making Go/No-go judgments. However, this task had a trade-off between judgment and movement accuracy. The participants must balance this trade-off in a limited time so as to enhance task performance. To assess judgment efficacy, participants also perform a hitting-only task without Go/No-go decision-making (NJ task) to compare to the hitting task with Go/No-go decision-making (J task)."*
- We designed this task because our interest is in rapid sensorimotor decision when the motor demands are high, such as in sports, and in hitting a target; the decision to hit or not to hit is important. We also elaborated our hypothesis in the Introduction (p.4, lines 95-98) as follows.
 - *"We hypothesize that the Go/No-go judgment is effective on task performance, but the efficacy is reduced in short time constraints because of a trade-off between judgment and movement accuracies. However, movement strategy is changed so as to compensate for the reduction in judgment efficacy."*

5. **Problem 4:** Line 96. What is d' ? as a judgement accuracy?

- **RESPONSE:** I apologize for the insufficient explanation. We have replaced "d" with "sensitivity d' , a statistical measure to quantify how a system distinguishes signal distribution from noise distribution" (p. 4, lines 116, 119-120) as follows. It indicates how well participants separate the correct signal distribution from noise distribution (in our task, Strike or Ball).

- *“Using hand movement profiles, we classified the observed movements into four possible **judgment** responses based on signal detection theory²⁹, i.e., Hit (Go in Strike), Miss (No-go in Strike), False Alarm (FA; Go in Ball), and Correct Rejection (CR; No-go in Ball) (see Supplementary Fig. 1a, b). As the judgment accuracy, we assessed **sensitivity d'**, a statistical measure to quantify how a system distinguishes signal distribution from noise distribution, for each TTC by calculating the ratio of four responses, Hit, Miss, FA, and CR.”*
6. **Problem 5:** From Suppl FIG 1B it looks as if all No-Go targets are in the plus-minus 15 degrees orientation. That would make the predictable. Does this carry any implication for their conclusions?
- **RESPONSE:** The targets were selected pseudo-randomly and the participants could not predict them. In our task, the Ball target was only ± 15 degrees, but in a real batting scenario, the variability of possible locations is large and some courses are more difficult than ours, so the decision-making process can be expected to be more difficult and severe.
7. **Problem 6:** Supl. FIG 1C-D, align these curves on movement onset, and make that the origin of time. They're hard to read just like that.
- **RESPONSE:** We apologize for the difficulty. We have added a new Supplementary Fig. 1c, e that aligns the data from Supplementary Fig. 1c-d (p.14, line 443) on movement onset, and made that the origin of time.
 - *“Using the Y direction acceleration data in J task, the judgment time for Hit responses was determined as the time when the value obtained by subtracting the average data of all No-go (CR and Miss) trials from each Hit response data exceeded the last time the threshold value was exceeded (Fig. 1c and Supplementary Fig. 1c, d, e).” (p.14, line 443).*
8. **Problem 7:** The statistics must provide accurate p-values. Providing boundaries ($p < 0.01$ or $p < 0.05$) is not good enough.
- **RESPONSE:** We have replaced boundaries with accurate p-values except description as $p < 0.0001$ when the p-value is less than 0.0001 (lines 130, 131, 133, 141, 143, 150, 171, 173, 174, 198) as follows.
 - *“A two-way analysis of variance (2-way ANOVA) showed significant main effects of TTC ($F(2, 66) = 35.22, p < 0.0001$) and task ($F(1, 66) = 29.90, p < 0.0001$); there was no significant interaction ($F(2, 66) = 3.58, p = 0.033$). The*

Post-hoc Bonferroni-corrected *t*-tests showed significant differences between tasks with 0.5 s ($p = 0.0078$) and 0.6 s TTC ($p = 0.00019$), but none at 0.4 s TTC ($p = 0.97$).” (lines 129-133).

- “We found a significant reduction in *the goal rate* (correct hitting rate to Strike trials) for J task at 0.4 s TTC compared with *the goal rate for J task* ($p = 0.010$, Fig. 2b), although there was no significant main effect of task ($F(1, 66) = 2.06$, $p = 0.156$) and significant interaction ($F(2, 66) = 5.49$, $p = 0.0063$),” (lines 140-143).
- “We found that judgment accuracy decreased significantly as TTC fell ($F(2, 33) = 39.77$, $p < 0.0001$), with a pronounced reduction from 0.5 s to 0.4 s TTC ($p < 0.0001$).” (lines 149-150).
- “although there was a significant difference ($F(1, 66) = 4.05$, $p = 0.048$) (Supplementary Fig. 3a). Indeed, we found a strong association between movement onset time and judgment time ($r^2=0.863$, $p < 0.0001$, Fig. 3a). In addition, movement onset time was significantly correlated with judgement accuracy ($r^2=0.372$, $p < 0.0001$, Fig. 3b).” (lines 170-174).
- “We found a significant correlation between the success rate of move-then-stop movements and judgement accuracy (d') at 0.4 TTC ($r^2=0.69$, $p = 0.0083$, Fig. 4c),” (lines 197-198).
- In addition, the exact description of the p-value in the figure has been added (p.5, lines 133-136, 144-148, 185) as follows.
 - “With regard to the task J success rate, there were significant differences between 0.4 s and 0.6 s TTC ($p < 0.0001$), and between 0.4 s and 0.5 s TTC ($p < 0.0001$). As for the NJ task success rate, there were significant differences between 0.4 s and 0.6 s TTC ($p = 0.0038$).” (lines 133-136).
 - “With regard to the goal rate for J task, the Post-hoc Bonferroni-corrected *t*-tests showed significant differences between 0.4 s and 0.6 s TTC ($p < 0.0001$), between 0.4 s and 0.5 s TTC ($p < 0.0001$), and between 0.5 s and 0.6 s TTC ($p = 0.038$). As for the goal rate for J task, there were significant differences between 0.4 s and 0.6 s TTC ($p = 0.023$).” (lines 144-148).
 - “we found that participants were apt to move their hand even in No-go trials as TTC fell ($F(2, 33) = 3.35$, $p = 0.047$, Fig. 4a, b).” (lines 184-185).

- Some of the F-values were wrong, so I have corrected them as follow (p.5, lines 130, 131). The suggestions from the results are not affected by this.
 - *“A two-way analysis of variance (2-way ANOVA) showed significant main effects of TTC ($F(2, 66) = 35.22, p < 0.0001$) and task ($F(1, 66) = 29.90, p < 0.0001$); there was no significant interaction ($F(2, 66) = 3.58, p = 0.033$).”*

9. **Problem 8:** There is a number of papers providing some interesting insight onto the SAT they mention. The authors should consider basing their argument on this previous work. From the top of my head Reynaud et al. 2020. They already start dealing with time constraints, as does as well the Thura et al. 2014 you cite.

- **RESPONSE:** Thank you for providing these important insights. I apologize for my lack of survey coverage. As you suggested, we have added this work, Reynaud et al. 2020 et al. in the Introduction (p.3, lines 55-59) and in the Discussion (p.7, lines 221-226) as follows.

- *“Reynaud et al.¹⁹ have revealed the trade-off between decision making and motor process according to time constraints of up to a few seconds by manipulating motor cost for a responding target in a token task. Their results indicate that fast and inaccurate decisions were often made before more demanding movements so that participants sacrificed decision making for action execution.”* in the Introduction (p.3, lines 55-59).
- *“Other studies have argued that urgent signals, assumed to be temporally strengthening signals related to the urgency in making a decision³⁶, increased the speed of triggering reaching movement and the vigor of saccades^{17,37}, suggesting a **shared** mechanism between decision-making and motor processes that involve the basal ganglia^{17,37,38}. A recent study on manipulating motor contexts showed that slow movements executed for demanding targets were often preceded by faster and less accurate decisions, indicating that participants sacrificed decision-making for action execution¹⁹. This suggests that decision urgency and movement vigor signals are likely to interact.”* in the Discussion (p.7, lines 221-226).

- We hope these revisions provide a more thorough discussion.

10. **Problem 9:** In the discussion, the authors mention that movement onset might force decision completion. The view that a decision can be finished before the movement has been completed cannot be qualified as embodied.

- **RESPONSE:** As you pointed out, the description that movement onset might force decision completion is inappropriate as embodied. We have revised to “*Our result suggests that movement and decision processes occurred concurrently*” (p. 8, lines 251-252).

11. **Problem 10:** The definition of success rate across the three task is not homogeneous, as you are quantifying different metrics across different experiments. I would be cautious to call all of this success rate on equal terms.

- **RESPONSE:** We agree that it was confusing to use the same word “*success rate*” for different metrics and have replaced the term “*success rate*” throughout the paper with “*NJ task success rate*”, “*J task success rate*”, and “*Goal rate for Strike targets*” to use more precise terms. To reflect this, we have revised the main figure (Fig. 1a, b) and Supplementary Fig. 2 and description in the Results (lines 121, 122-124, 126-128, 133-136, 140, 141, 144-147, 151, 152) and in the Data analysis (lines 429, 430, 432-436, 438,439) and in the caption of Fig. 2 (p.21, lines 674-676, 678, 680) as follows.

- “The *Hit responses* were further divided into two responses according to goal success, e.g., *Hit-Goal* and *Hit-No Goal*. We assessed *goal rates* (correct hitting; *Hit-Goal rate to all Strike trials*) and overall task success rate for J task, the *J task success rate*, which was defined as the ratio of the sum of the correct hitting (*Hit-Goal*) and the correct judgment (CR) responses in all trials (see Methods). Corresponding task success rates for NJ task, *NJ task success rate*, were also evaluated as the ratio of the correct hitting.

Both the J task success rate and the NJ task success rate decreased with shorter TTCs, but the trends differed between tasks (Fig. 2a).” (lines 121-129).

- “*With regard to the task J success rate, there were significant differences between 0.4 s and 0.6 s TTC ($p < 0.0001$), and between 0.4 s and 0.5 s TTC ($p < 0.0001$). As for the NJ task success rate, there were significant differences between 0.4 s and 0.6 s TTC ($p = 0.0038$).*” (lines 133-136).
- “*We found a significant reduction in the goal rate (correct hitting rate to Strike trials) for J task at 0.4 s TTC compared with the goal rate for J task ($p = 0.010$, Fig. 2b),*” (lines 140-142).
- “*With regard to the goal rate for J task, the Post-hoc Bonferroni-corrected t-tests showed significant differences between 0.4 s and 0.6 s TTC ($p <$*

0.0001), between 0.4 s and 0.5 s TTC ($p < 0.0001$), and between 0.5 s and 0.6 s TTC ($p = 0.038$). As for the goal rate for J task, there were significant differences between 0.4 s and 0.6 s TTC ($p = 0.023$).” (lines 144-148).

- “The reduction in *the J task success rate* was mainly caused by lower judgement accuracy, since *the NJ task success* barely changed from 0.5 to 0.4 TTC (Fig. 2a).” (lines 151-152).
- “We defined *the J task success rate and the NJ task success rate, which regarded as total task performance, and the goal rate for Strike targets*, as follows (see also Supplementary Fig. 1b).

$$J \text{ task success rate (Strike + Ball)} = \frac{\text{Hit-Goal} + \text{CR}}{\text{Total}}$$

$$NJ \text{ task success rate (Strike + Ball)} = \frac{\text{Goal}}{\text{Total}}$$

$$goal \text{ rate (Strike only)} = \frac{\text{Strike-Goal}}{\text{Strike}}$$

The *J task success rate* was defined as the number of Hit-Goal and CR responses divided by the total number of trials, and *the NJ task success rate* was defined as the number of goal responses divided by the total number of trials. In order to examine the impact of judgment on hitting accuracy, we determined the *goal rate (Strike)*. It was defined as the number of *Strike-Goal* responses divided by the number of *Strike trials* for J task *and* NJ task.” in the Data analysis (lines 429-439).

- “**a, b** The mean *task success rate* for Strike and Ball targets (**a**) and *the mean goal rate* for only Strike targets (**b**) for each TTC are plotted as orange lines (*the J task success rate*) and blue lines (*the NJ task success rate*) (Mean \pm SEM, * $p < 0.05$, ** $p < 0.01$). **c** The mean judgement accuracy (*d'*) for each TTC is shown by orange lines (Mean \pm SEM, * $p < 0.05$, ** $p < 0.01$) with individual data points overlaid. Please note that *the J task success rate and judgment accuracy* decreased as TTCs were shortened, and judgment was ineffective in terms of *J task success rate* if TTC was less than 0.5 seconds.” in the caption of Fig. 2 (p.21, lines 673-680).

In addition to the pointers, we have corrected the wording to be more appropriate and fixed wrong words and variable values in Supplementary Information.